# Replicate sequencing libraries are important for quantification of allelic imbalance

Asia Mendelevich [1,2,8✉], Svetlana Vinogradova [2,8], Saumya Gupta [2,3], Andrey A. Mironov[4,5], Shamil R. Sunyaev[6,7] & Alexander A. Gimelbrant [2,3✉]

A sensitive approach to quantitative analysis of transcriptional regulation in diploid organisms is analysis of allelic imbalance (AI) in RNA sequencing (RNA-seq) data. A near-universal practice in such studies is to prepare and sequence only one library per RNA sample. We present theoretical and experimental evidence that data from a single RNA-seq library is insufficient for reliable quantification of the contribution of technical noise to the observed AI signal; consequently, reliance on one-replicate experimental design can lead to unaccounted-for variation in error rates in allele-specific analysis. We develop a computational approach, Qllelic, that accurately accounts for technical noise by making use of replicate RNA-seq libraries. Testing on new and existing datasets shows that application of Qllelic greatly decreases false positive rate in allele-specific analysis while conserving appropriate signal, and thus greatly improves reproducibility of AI estimates. We explore sources of technical overdispersion in observed AI signal and conclude by discussing design of RNA-seq studies addressing two biologically important questions: quantification of transcriptome-wide AI in one sample, and differential analysis of allele-specific expression between samples.

[1] Skolkovo Institute of Science and Technology, Moscow, Russia. [2] Center for Cancer Systems Biology, Dana-Farber Cancer Institute, Harvard Medical School, Boston, USA. [3] Broad Institute of Harvard and MIT, Cambridge, USA. [4] Lomonosov Moscow State University, Faculty of Bioengineering and Bioinformatics, Moscow, Russia. [5] Institute of Information Transmission Problems, Russian Academy of Sciences, Moscow, Russia. [6] Department of Biomedical Informatics, Harvard Medical School, Boston, USA. [7] Division of Genetics, Brigham and Women's Hospital, Boston, USA. [8] These authors contributed equally: Asia Mendelevich, Svetlana Vinogradova. ✉email: a.mendelevich@skoltech.ru; gimelbrant+lab@gmail.com

RNA sequencing (RNA-seq) is a widely used technology for measuring RNA abundance across the whole transcriptome[1]. In samples from humans and other diploid organisms, comparison of the activity of the maternal and paternal alleles is an especially informative approach to understanding regulation of gene expression. The two parental copies of a gene share the trans-regulatory environment of the same nucleus, making allelic imbalance (AI) in expression specifically reflective of the cis-regulatory mechanisms[2,3]. This has led to use of AI analysis in a growing number of studies of regulatory variation[4–8]. In addition to genetic mechanisms, allele-specific analysis of the transcriptome can reveal epigenetic gene regulation in cis, usually associated with either imprinting[9], X-chromosome inactivation[10], or autosomal monoallelic expression[11–15].

For accurate quantitative analysis of RNA-seq, it is necessary to separate biological signal from technical variation (noise). A nearly universal practice in RNA-seq studies of AI is to use one technical replicate, i.e., generate a single sequencing library per sample. We found only two publicly available datasets with technical replicates; one study produced seven RNA-seq libraries from each of five human cell lines[16] and the other, two technical replicates from one human cell line[17]. In experiments without technical replicates, accounting for technical noise relies on assumptions about its properties. The de facto standard is the simple binomial test with correction for multiple hypotheses[5,18–20]; there are also methods that incorporate over-dispersion terms into one-replicate analysis[21,22]. For analysis of biological replicates, pooling or averaging data is usually employed (e.g.,[17,23]).

We asked if a single technical replicate provides sufficient information to quantify contribution of technical noise to the observed AI signal. Our analyses show that it is insufficient, unless very restrictive assumptions are made about the true underlying AI signal or the exact characteristics of the noise.

Furthermore, we performed an experimental assessment of technical noise in AI analysis of RNA-seq data, by producing multiple replicate libraries from the same RNA, varying methods of library construction and amount of RNA input. In these and publicly available datasets, we found that noise can greatly vary between experiments, and thus its properties cannot be assumed to be uniform.

In this work, we devise an approach to estimate technical noise based on the comparison of two or more replicate RNA-seq libraries, and implement it in a software tool, Qllelic (github.com/gimelbrantlab/Qllelic). Its performance favorably compares both with the simple binomial test with correction for multiple hypothesis testing[5,18–20], and methods that incorporate over-dispersion terms into one-replicate analysis[21,22]. It also performs better on data with biological replicates than pooling or averaging, although there are limitations involved in the use of biological replicates without technical ones. Finally, as a practical guide, we outline typical use cases for application of Qllelic for allele-specific analysis of RNA-seq data.

## Results

**One technical replicate in RNA-seq is insufficient for estimation of AI technical noise.** For accurate analysis of RNA sequencing data, biological signal should be separated from the experimental noise. One obvious source of technical variation is sampling due to limited sequencing depth; this variation can usually be accounted for using binomial distribution[5,24]. Many existing approaches to AI analysis also incorporate an additional component of noise, extra-binomial overdispersion[21,25,26].

In some highly stylized models of the underlying "true" distribution of AI signal, a single RNA-seq library can provide sufficient information for separating noise and AI signal. An example of such a model is a discrete distribution of AI values, such that all transcripts are either exactly 50:50 biallelic or completely biased towards one of the alleles (see Supplementary Note S1). More realistic prior models of true distribution of AI signal and noise are called for when addressing more quantitative questions, such as differential AI analysis, rather than classification of genes into discrete categories like biased/unbiased. For the model of the true AI signal, a natural extension of the discrete trimodal distribution is making it continuous. The resulting bell-shaped distribution with fat tails resembles a typical observed distribution of AI values in transcriptome analysis (see Supplementary Fig S13). Such distribution of true AI signal can be modeled as a beta-binomial distribution. Technical noise is often modeled as a beta-binomial distribution incorporating both sampling noise and overdispersion[21,22,25,27].

Even under this simple continuous model, an observed distribution of AI values could result from different parametrizations of the beta-binomial distributions describing true AI signal and noise (Fig. 1a). Simulation analyses (Supplementary Note S1) show that such parametrizations, assigning very different amounts of noise (Fig. 1a), cannot be distinguished by the Mann–Whitney–Wilcoxon and Kolmogorov–Smirnov tests (Supplementary Note S1). Here, noise incorporates all sources including technical noise from library preparation and measurement, and biological variation.

For two additional classes of models, we analytically show (Supplementary Note S1) that multiple parametrizations may result in observationally equivalent distributions. In one such class, both true signal and noise are normally distributed. In another, noise is described by either beta-binomial or binomial distribution, and true AI values are described as a mixture of three Dirac delta functions or Beta mixture, respectively.

These observations strongly suggest that, unless models are very restrictive, RNA-seq data from a single technical replicate is insufficient to achieve definitive parametrization of technical noise and true AI signal.

In the rest of the paper, we describe and experimentally test a method to account for technical noise and accurately estimate AI using RNA-seq data from two or more technical replicates.

**Datasets used.** There are two principal variables involved in the generation of RNA sequencing libraries: (a) different protocols can be used; and (b) with a given protocol, library preparation can start with different amounts of RNA. To probe these two variables in a compact way, we generated three sets of poly-A enriched RNA-seq libraries from the same total RNA extracted from the mouse kidney. To increase the fraction of allele-specific reads, we used RNA from a polymorphic F1 mouse cross (129S1 × Cast/Ei) with genome-wide density of single nucleotide polymorphisms (SNPs) of ~1/118 bp, about 10-fold higher than in humans.

Each set ("experiment") consisted of six libraries prepared in parallel. Libraries for experiment 1 ("NEBNext (100ng)") were prepared using a protocol for large amounts of input RNA (100 ng of total RNA, see details in Methods). Experiments 2 and 3 featured libraries prepared using SMART-Seq v4 Ultra Low Input RNA Kit (Clontech) with amounts of input total RNA bracketing the recommended range—10 ng and 100 pg, respectively ("SMARTseq (10ng)" and "SMARTseq (0.1ng)"). Information on these sequencing data is summarized in Supplementary Table S1.

We also analyzed published datasets. Information on these is summarized in Supplementary Table S1, with further details on their analysis listed in Supplementary Table S2 (human samples[16]) and Supplementary Table S3 (mouse samples[28]).

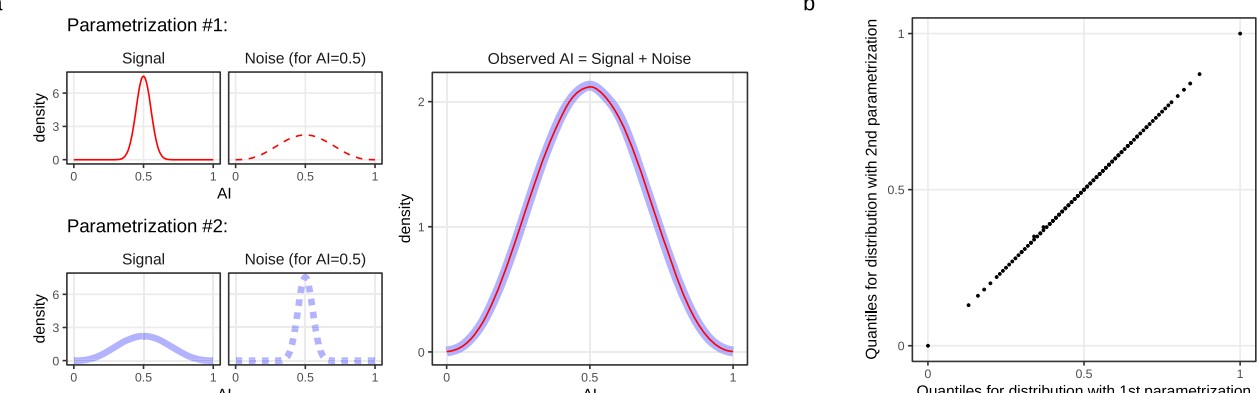

**Fig. 1 Different combinations of signal and noise parameters result in indistinguishable observed distributions of AI values. a** Two simulated parametrizations (left) of true AI signal ($AI_{true}$; solid line) and noise (dashed line) that combine to produce overlapping observed AI values (right; red and blue, respectively). These and similar observations are indistinguishable by Mann–Whitney–Wilcoxon and Kolmogorov–Smirnov tests; see Supplementary Note S1. AI distributions are shown for allelic coverage 100. Noise distribution shown at $AI_{true} = 0.5$. Both signal and noise are modeled using beta-binomial distributions; the following parameters are shown: [$\rho_{Signal} = 0.001$, $\rho_{Noise} = 0.1$] and [$\rho_{Signal} = 0.1$, $\rho_{Noise} = 0.001$]; simulation sample size 500,000. Other coverage levels and combinations of $\rho$ values are shown in Supplementary Note 1. **b** Quantile–Quantile (QQ)-plot for distributions set by parametrizations 1 and 2 from panel (**a**). Quantiles were taken from 0 to 1, with step 0.01.

**Variation in the AI estimates across replicate RNA-seq libraries**. To assess the consistency of AI estimates, we applied a uniform statistical test across technical replicates and experiments to construct an "AI discordance map" for replicate pairs (Fig. 2a). For each gene in either of the replicates being compared, we tested the null hypothesis of a perfectly biallelic expression ($H_0$ of AI = 0.5) using a binomial test. Genes with the $H_0$ rejected (with $P = 0.05$, after the conservative Bonferroni correction for multiple hypothesis testing[29]) are classified as having AI bias; the rest are classified as showing no bias. Only the genes with discordant calls in the two replicates are shown on the resulting map, with a separate highlight for genes showing the opposite bias in the two replicates. This statistical procedure, widely used in studies of AI[5,18–20] yields a large number of discordant calls between technical replicates (Fig. 2b). Note that the discordant genes as a whole are not limited to the boundary determined by the binomial test, suggesting that the pure binomial noise is not a close fit to the experimentally observed dispersion of AI values (see Supplementary Note S2).

It is known that in RNA-seq there is "overdispersion"—more signal dispersion than would be expected from an assumption of binomial noise[30–32]. To account for overdispersion, some within-replicate analysis is suggested (approaches include comparing sets of reads sampled without replacement from the same read pool[17] or bootstrapping[30]). Indeed, sampling without replacement (which is equivalent to sequencing two aliquots from the same library) is more discordant than binomial sampling within one replicate (Supplementary Fig. S1), suggesting that this procedure captures some overdispersion. However, distinct replicates within the same experiment are much more discordant than sampling without replacement shows (Fig. 2c), and they reveal additional noise not detectable by analysis within a single library.

Importantly, the concordance dramatically differed across experiments (Fig. 2b, left to right), while similar for pairs of replicates within the same experiment: 49.2% ± 4.7 (s.d.) for experiment 1 (52.3% ± 1.3 when one outlier replicate was removed), 61.8% ± 0.6 for experiment 2 and 38.3% ± 1.0 for experiment 3 (Fig. 2e, left). RNA abundance values (non allele-specific) appear to agree much better between replicates, even only counting reads covering SNPs (Supplementary Fig. S2); greater variation in AI values can be ascribed to these values being proportions and thus amplify small variations. In addition to our datasets, we assessed publicly available data, showing variability in overdispersion in those experiments[16,28] (Supplementary Tables S2 and S3). When multiple replicates were available, we also observed that the overdispersion assessed in pairs of replicates was similar for all replicates within an experiment.

We thus conclude that AI overdispersion observed in RNA-seq data is experiment-specific.

**Estimation of AI overdispersion from observed and modeled data**. In order to quantify the experiment-specific overdispersion between a pair of replicate libraries, we assess how its experimentally observed value differs from the expected value in a fitted model. Rather than fit parameters for all levels of coverage using a combination of negative binomial and beta-binomial distributions (as in[21,25–27,33]), we bin genes by total allelic coverage and use this discretized model for further analysis (Fig. 3).

In analyzing each bin, we use a continuous distribution of AI values between the extreme allelic biases, rather than a trimodal classification (AI = {0, 0.5, 1}) typically used in beta-binomial models of allelic overdispersion[25,26,33]. To fit the experimentally observed AI distribution, we use a mixture beta distribution, which should be a better fit for an AI distribution with heavy tails (Fig. 3a). To gauge the experimentally observed dispersion, we perform quantile analysis of the distribution of ΔAI values within the coverage bins, where ΔAI is a difference between two replicates in AI values for a gene (Fig. 3b).

To estimate the overdispersion, the observed quantile values are normalized on the expected ones. Note that genes with different AI have different impacts on the overall signal dispersion (see Supplementary Note S3). Thus, the distribution of AI in each specific bin should be accommodated in the model.

To model the expected distribution of ΔAI in each coverage bin and compute the corresponding quantiles, we perform the following procedure. In each bin, actual distribution of AI for genes is fit using a beta-binomial mixture model (Fig. 3c, top). Using fitted parameters from that model, we then simulate two RNA-seq replicates (Fig. 3c, middle). The expected distribution of ΔAI comes from an assumption of binomial sampling of alleles in these two simulated replicates. Finally, we calculate the quantiles

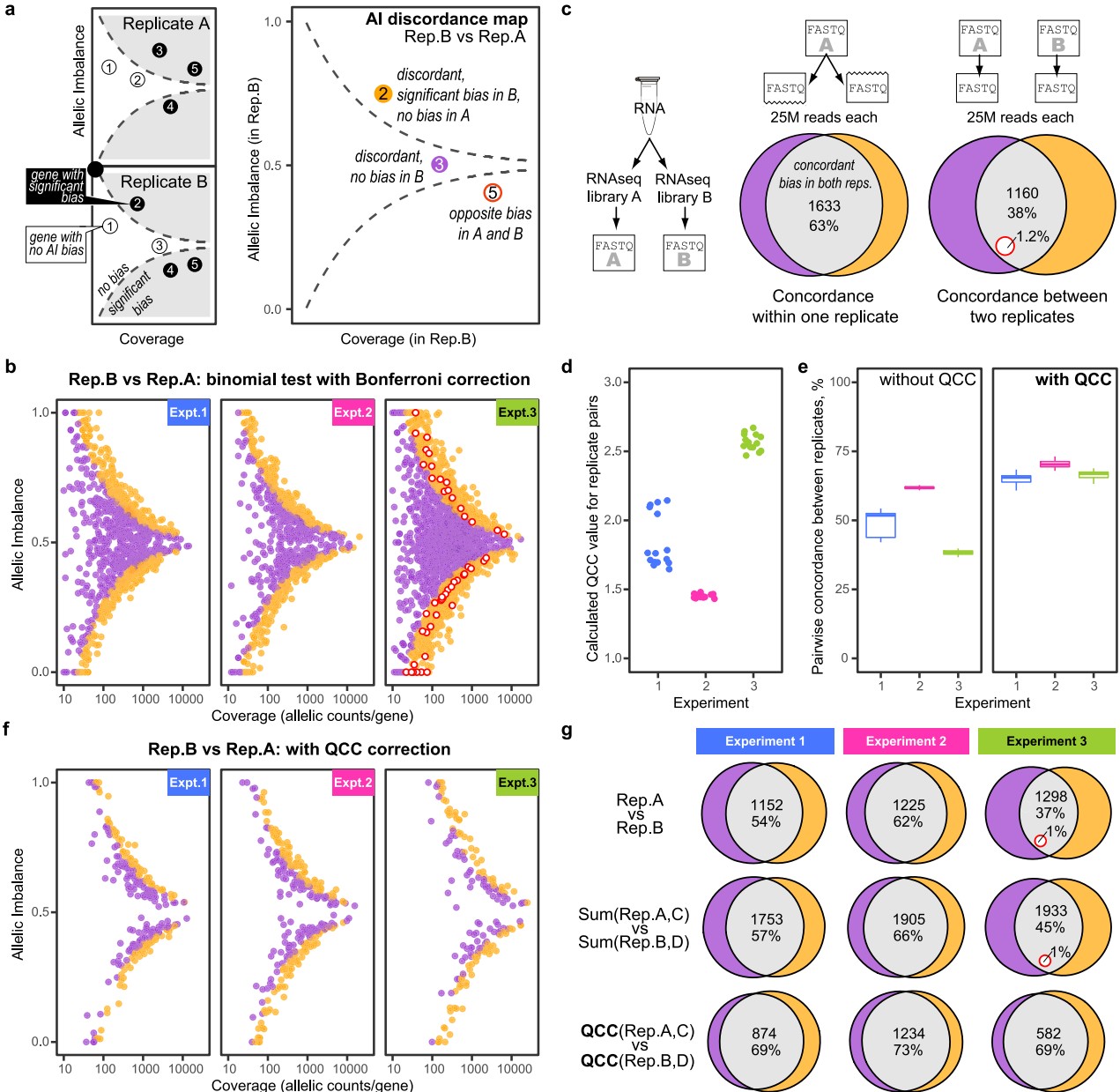

**Fig. 2 Allelic imbalance values vary between technical replicate libraries and RNA-seq experiments. a** Explanation of AI discordance map. Type of discordance: orange–no bias in RNA-seq replicate A, bias in B; purple--no bias in B, bias in A; red/white circle--bias in both, in opposite directions. **b** AI discordance maps for representative pairs of technical replicates for three different experiments, all prepared from the same total RNA (see Methods and Supplementary Table S1). Significant bias: $H_0$ of AI = 0.5 for each gene rejected by two-sided binomial test at $p = 0.05$ with Bonferroni correction. Colors as in (**a**). All comparisons, here and elsewhere, sample 30M uniquely mapped reads per replicate, unless noted otherwise. **c** Replicate RNA-seq libraries prepared from the same RNA compared to subsets of reads from the same library. Euler diagrams show genes with significant AI bias. Colors as in (**a**); percentages show fraction of all discordant genes. Data: RNA-seq from 129 × CastF1 mouse, replicates 1 and 2 from Experiment 3 (see Supplementary Table S1). **d** Quality Correction Coefficient (QCC), a measure of AI overdispersion defined in this work, calculated (see Fig. 3) for all 15 pairs of replicates within each of Experiments 1 (blue), 2 (red), or 3 (green). Notice general consistency of QCC values within experiments, and sensitivity to one outlier replicate in Experiment 1. **e** Fraction of concordantly biased genes [cf. grey area in (**c**)] for all 15 pairs of replicates within Experiments 1-3. Left: two-sided binomial test. Right: two-sided proportional test with QCC correction. Boxplot elements–center line: median; box: upper and lower quartiles; whiskers: 1.5 x interquartile range; points: outliers. **f** Same as (**b**), except $H_0$ tested using proportional test with QCC correction. **g** Application of QCC increases concordance between replicates and between experiments. Colors as in (**c**). Top row: comparison of two individual replicates [replicates 2 and 3; Supplementary Table S1], 30M reads each; $H_0$ test: two-sided binomial with Bonferroni correction. Middle and bottom rows: comparison of pooled pairs of replicates [replicates 2 + 4 vs 3 + 5; Supplementary Table S1], 30M reads per replicate. Middle: two-sided binomial test with Bonferroni correction. Bottom: two-sided proportional test with QCC correction.

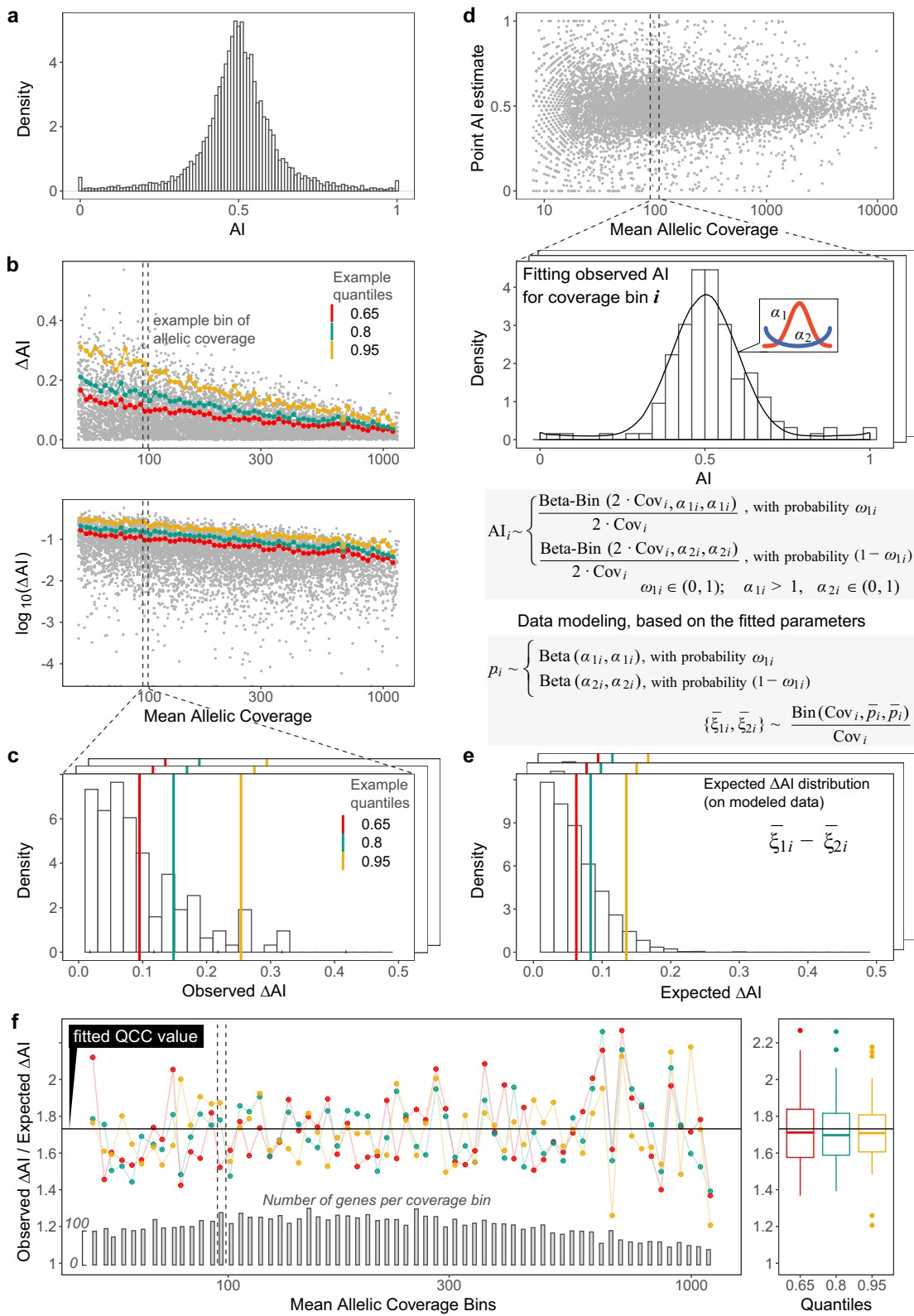

for the expected ΔAI distribution (Fig. 3c, bottom), and find the ratio of the observed to expected quantiles.

This ratio of observed to expected ΔAI quantiles appears to be a constant, with some random fluctuations (for the two replicates shown in Fig. 3d, this ratio is 1.73 ± 0.18). This

constant depends on the experiment. Idealized Poisson sampling corresponds to no overdispersion and the constant value of 1; in experimental observations we expect this value to be ≥1. We call this fitted experiment-specific quantity the Quality Correction Coefficient (QCC).

**Fig. 3 Derivation of Quality Correction Coefficient from observed and modeled AI differences between technical replicates. a–c** Finding observed distributions of gene-level AI differences between replicates (ΔAI). **a** Distribution of point AI estimates for genes with allelic coverage over 10 in six pooled replicates (180M reads total) of Experiment 2. **b** Calculation of the observed distributions of AI differences between two replicates. After sampling an equal number of reads from two technical replicates, AI is calculated for each gene. Plotted is ΔAI against mean SNP coverage in linear (top) or log (bottom) scale. Genes are binned by log coverage (an example bin is shown). Quantiles are calculated per bin. Here and elsewhere, three example quantiles are shown: 0.65 (red), 0.8 (green), 0.95 (orange). **c** Distribution of the observed ΔAI values in a bin. Example quantiles are shown. **d–e** Calculation of the expected distributions of AI differences between replicates. **d** Top: AI for each gene is calculated after pooling SNP counts from both replicates. Note that we use mean SNP coverage, so the bins contain the same genes in both replicates. Bottom: For each coverage bin, distribution of AI values is fitted with a mixture of two symmetric beta-binomial distributions (red and blue curves). **e** Distribution of the expected ΔAI values in a bin. To generate expected ΔAI: we generate a simulated sample of 5000 genes, with the distribution of exact allelic imbalance values ($\xi$) according to the fitted parameters; from these genes, we then simulate two replicate datasets, with SNP coverage according to the bin, and sampling from binomial distribution; finally, we calculate the simulated ΔAI for each gene and find quantiles for their distribution. **f** Ratios of observed and expected values for the example ΔAI quantiles. Fitted black line defines Quality Correction Coefficient. Boxplots (right) summarize values for all coverage bins (left). Boxplot elements (right)--center line: median; box: upper and lower quartiles; whiskers: 1.5 x interquartile range; points: outliers.

**Application of QCC increases concordance between replicates**. Once the QCC value is calculated for a pair of replicates, it is straightforward to use it to correct for the extra-binomial variation. To account for the widening of AI distribution, allelic counts are divided by QCC$^2$ and used as input for proportional test, which allows us to perform the analysis on non-integer values (see Methods for details). All the "QCC-corrected" analyses described below also incorporate Bonferroni correction for all analyzed genes to account for multiple hypothesis testing.

QCC values reflect the "quality" of the data, in the sense of concordance of AI calls between replicates: experiments with lower concordance lead to higher QCC values (cf. Fig. 2d with Fig. 2b and e). QCC values are similar for all pairs of replicates within the experiment (Fig. 2d), strongly suggesting that QCC reflects an experiment-specific property.

QCC correction leads to increased concordance between replicates within each experiment and between experiments (Fig. 2e), indicating that QCC accounts for much of the experiment-specific overdispersion. Note that one of 6 replicates in Experiment 1 appears to be an outlier, with higher QCC values when compared with all other replicates. After QCC correction this difference in concordance is greatly diminished.

Notice that after the QCC correction, the discordant calls are much fewer in number, and these numbers are much closer to each other across experiments (Fig. 2f). Furthermore, these discordant calls are distributed close to the boundary between significant and insignificant bias determined by the QCC-corrected statistical test (cf. Figs. 2f and 1b), suggesting a better agreement of noise expectations with the observed data (see Supplementary Note S2). QCC correction reduced the number of genes called as imbalanced, while greatly increasing concordance between the pairs of replicate libraries (Fig. 2g). More noisy datasets (e.g., Experiment 3) showed greater reduction in the number of genes called imbalanced (Fig. 2g and Supplementary Fig. S3c), as should be expected when the confidence in these data is (appropriately) lowered.

Note that we apply QCC correction after pooling SNP counts from both replicates, to take advantage of the available data. Importantly, the improvement in the concordance level is due primarily to the use of the QCC correction computed from two technical replicates, rather than the fact of pooling the data from replicates (Fig. 2g).

**Application of QCC improves differential AI analysis**. All analyses we discussed so far focused on a binary classification of genes into biased or unbiased, depending on testing of the $H_0$ of AI = 0.5. A more informative question is the quantitative description of AI for a gene (or another region of interest), which includes its AI point estimate and confidence interval (CI) for the true proportion. Accounting for the sample-specific AI over-dispersion enables more accurate differential analysis of AI when comparing two or more samples.

Our dataset allows us to provide definitive analysis of false positive (FP) rates: since we know the data came from exactly the same RNA, we expect zero differential calls after correction on multiple testing, so any differences deemed significant are FP. With analysis done under binomial assumptions (with Bonferroni correction for multiple testing), comparisons across replicate RNA-seq libraries and across experiments show hundreds of genes with significant differential AI (Fig. 4a, left). This is also the case when applying some existing tools on the same data[21,22] (Supplementary Fig. S4). By contrast, QCC correction completely removes such false positives from within-experiment comparisons (Fig. 4a, right). The number of false positives in across-experiment comparisons using QCC correction is dramatically decreased but not reduced to zero (Fig. 4a). This suggests that QCC-corrected AI values can be used to compare AI across experiments, but systematic technical differences in protocols can still lead to some FP calls.

We also asked to what extent QCC correction is a better fit to the observed error distribution than the binomial assumption. We calculated the number of FP calls by testing whether point AI estimates from six pooled replicates are contained within the CI of the AI estimates calculated in the three following ways. Fig. 4b shows the FP rates for CIs obtained from one replicate under binomial assumption (left); pooled pairs of replicates under binomial assumption (middle); and QCC-corrected from pairs of replicates (right). The expectation (see Supplementary Note S4) is that after the Bonferroni correction, there should be close to zero FPs. Among tested approaches, only the QCC correction greatly decreased FP. Other tools also showed a large number of FP calls in this analysis (Supplementary Fig. S4b, c).

We then asked whether this decrease in FPs was due to overly conservative noise correction. Fig. 4c, d shows that the computed QCC value is near the point where FP rate reaches the plateau of 0, but not much higher. This suggests that the precision/recall tradeoff is close to optimal (see also Supplementary Fig. S11d). The tradeoff between false positives and signal in differential AI analysis is explored in detail in Supplementary Note S5. In that Note, we perform power analysis and show that any cost when using QCC-corrected test can be compensated with additional sequencing. By contrast, cost in false positives from lack of adjustment for technical overdispersion cannot be reduced by additional coverage.

We conclude that for differential AI analysis, lack of overdispersion correction is likely to result in a very large number of false positives, while QCC correction removes these

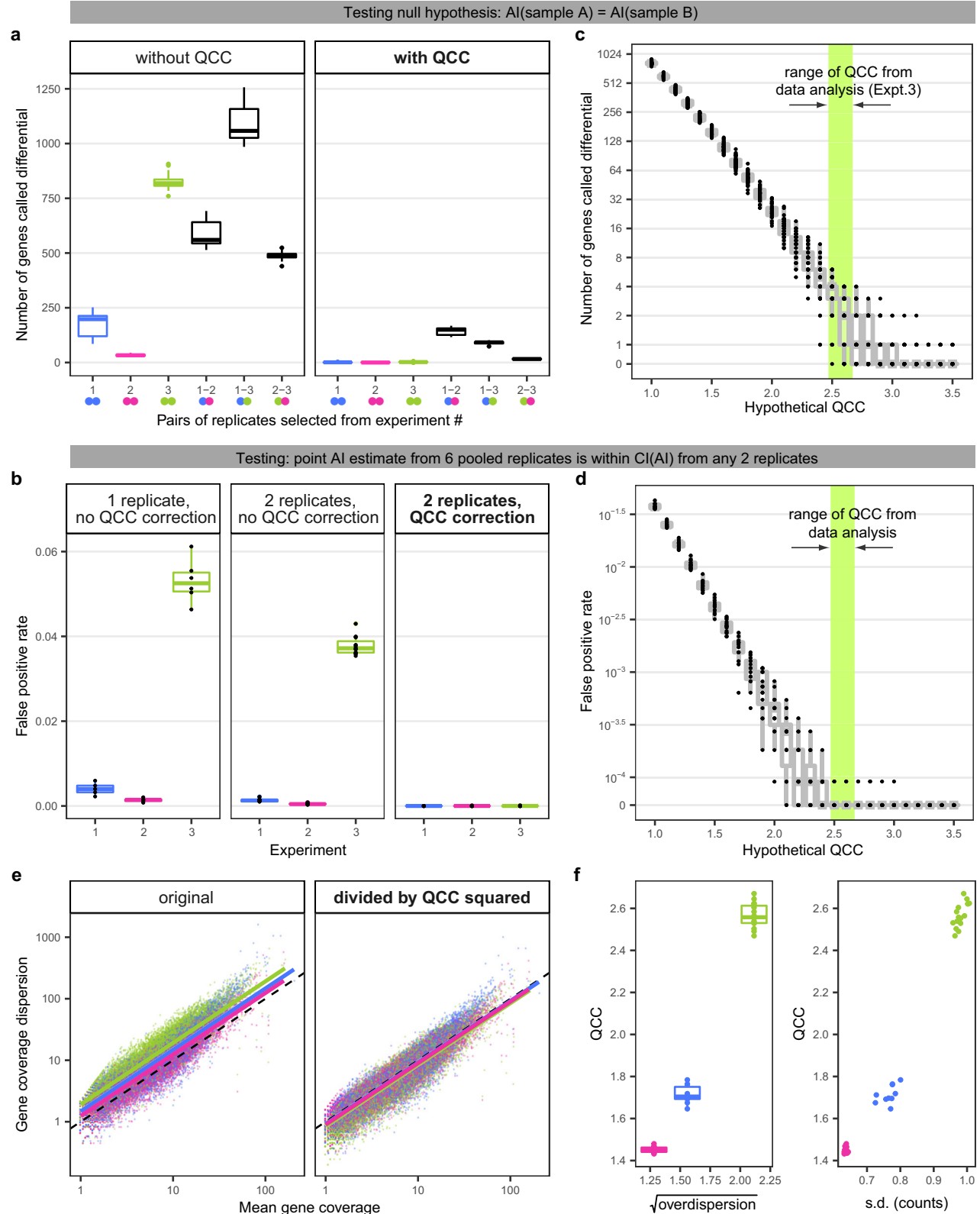

very effectively. Our analyses of the real and simulated data also suggest that QCC correction is not dramatically overconservative.

**Sources of AI overdispersion: data analysis**. To identify possible sources of AI overdispersion, we considered different stages of an RNA-seq experiment (Supplementary Fig. S5a): (1) steps from the

biological object up to and including RNA isolation; (2) generation of sequencing library from RNA; (3) the library sequencing process itself; (4) processing of sequencing data, from read alignment to statistical analysis of allelic imbalance.

Contributions from step 1 were excluded from our experiments by design: all 18 replicate libraries were prepared from the same mouse kidney total RNA. Taking bulk aliquots of purified RNA

**Fig. 4 QCC enables differential AI analysis and is correlated with abundance overdispersion. a** Number of genes with apparent differential AI in the same biological sample (false positives) before (left) and after QCC correction (right). Within each of three experiments, all 45 pairwise comparisons were performed. For comparisons between experiments, 45 sets of pairs were randomly chosen from 225 possible combinations. QCC values were calculated for each pair, and genes with significantly different AI in each set were identified. Experiments : 1 (blue), 2 (red), or 3 (green). Boxplot elements everywhere as in Fig. 2. Two-sided test was used here and in (**b**), (**c**), (**d**). **b** Impact of QCC on false positive (FP) rate. FPs are defined as genes for which the point AI estimate from all pooled replicates is not within CI from one replicate (left), two pooled replicates (center), and two replicates with QCC correction (right). Outlier from Experiment 1 (replicate 1) was removed from this analysis. All individual datapoints are shown in black. **c**, **d** Calculated QCC value (contained within the color bar) are close to optimal balance between FP and signal. **c** Number of FP genes (differential expression AI in the same biological sample) calculated for different potential values of QCC for all 45 possible combinations of two pairs of replicates from Experiment 3. **d** FP rate (as in (**b**)) for different possible QCC values calculated for all possible pairs of replicates in Experiment 3. **a–d** Note that the unit of comparison is composed of two technical replicates in both the binomial test and the QCC test. **e** Differences between experiments in abundance overdispersion are proportionate to QCC. Left: Abundance overdispersion for each experiment can be fitted as log-linear regression (solid lines) above expected Poisson dispersion (dotted line). Right: same after overdispersion was divided by QCC2. The outlier replicate from Experiment 1 was removed for (**e**) and (**f**). **f** Correlation of QCC and abundance overdispersion. QCC values are the same as in (**e**). Abundance overdispersion for each experiment calculated as exponent of intercept of log-linear regression (see (**e**)) between mean and dispersion of total counts: Left: for all replicates in an experiment; Right: for all possible pairs of replicates.

can be considered a fair Poisson sampling process (unlike in single-cell experiments, where there are additional sources of noise such as transcription bursts[34,35]).

Data analysis (step 4) includes multiple sub-steps, and we can assess their contributions to AI overdispersion (Supplementary Fig. S5b, c, Supplementary Fig. S4). First, we note that these steps taken together are not a major contributing factor to variability between experiments, since input of identical data results in consistent AI and QCC values (*modulo* noise from simulation procedures).

Note that the procedures used for allele counting can by themselves contribute to AI overdispersion. For example, several popular tools for RNA-seq analysis (including Kallisto[36], Salmon[37] and RSEM[38]) use not just the reads that overlap SNPs, but allow distribution of the rest of the reads between the two alleles. Such haplotype assignment should lead to a linear increase in coverage but a quadratic increase in standard deviation, and thus an increase in overdispersion. Application of one such tool[36] illustrates the systematically higher overdispersion compared to the use of SNP-overlapping reads only (Supplementary Fig. S6).

We assessed the contribution of the QCC calculation procedure to estimated AI overdispersion. On simulated total allele counts with known overdispersion, the QCC values were as expected (Supplementary Fig. S7; denoted as i in Supplementary Fig. S5c), suggesting that the QCC calculation by itself contributes little to the noise. A related analysis starting with random binomial sampling from one replicate's sequencing data (ii in Supplementary Fig. S5c) should show only overdispersion related to the allele counting and QCC calculation process. It yielded QCC values of 1.01–1.04 (Supplementary Fig. S8b), i.e., also close to no overdispersion (QCC ~ 1.0).

Note that when we randomly divide paired reads from the same run into two equal parts (cf. Fig. 2c, center), these "half-replicates" are not in a binomial relationship with each other. In these comparisons (iii in Supplementary Fig. S5c), QCC values ranged from 1.45 to 1.48 (Supplementary Fig. S8), reflecting the dispersion that came in the data from one sequencing run of a single library.

It is well documented that (non-allele-specific) counts in RNA-seq show extra-binomial overdispersion[30–32]. We asked how this "abundance overdispersion" is related to the AI overdispersion quantified by QCC. Abundance overdispersion can be seen in all three experiments: the log-linear fit lines are above the expected Poisson dispersion (Fig. 4d, left). Moreover, overdispersion was different for the three experiments. Strikingly, when dispersion values for each gene were divided by QCC squared, the regression lines for all experiments nearly coincided with each other and with the Poisson expectation (Fig. 4d, right). Accordingly,

abundance overdispersion was correlated with QCC values (Fig. 4e). In simulations, QCC was also very strongly correlated with set overdispersion values (Supplementary Fig. S7). Based on these analyses, we hypothesize that abundance overdispersion and AI overdispersion result from largely the same processes. While the correlation between abundance overdispersion and AI overdispersion in these examples was strong, calculation of the QCC correction is a more robust procedure that does not depend on this correlation holding for all experiments and any data processing procedure.

**Sources of AI overdispersion: experimental**. Analysis of two sequencing runs with the same library (iv in Supplementary Fig. S5c) yielded QCC values similar to in silico sampling from within the same library (Supplementary Fig. S8), suggesting that an additional sequencing run is similar to having more reads in the original run (compare ii and iv in Supplementary Fig. S5c). This is consistent with only a small contribution, if any, from the sequencing process to variation of AI overdispersion between replicates. For a more certain conclusion, a greater number of experiments would be needed.

QCC was much greater for between-replicate comparisons than for half-replicate comparisons (v in Supplementary Fig. S5c), showing that there is additional noise coming from each replicate. Note that this underscores the point that analysis within a single replicate does not allow one to correctly account for overdispersion.

With biological noise (step 1) excluded by design and steps 3 and 4 eliminated, only library preparation (step 2) remains as a major source of the AI overdispersion. This is reinforced by the observation that different procedures (overall protocols or adjustments made for different starting amounts of RNA) in our Experiments 1–3 produced libraries with greatly varied QCC values, while technical replicates within each experiment are similar to each other (see Fig. 2d).

Generation of RNA-seq libraries involves multiple steps, from reverse transcription and cDNA fragmentation to library amplification, and these steps can substantially vary between protocols. Detailed analysis of specific protocols is outside the scope of this work. However, one common concern in deep sequencing experiments is the impact of PCR amplification artifacts[39–41].

To assess the impact of amplification artifacts on AI overdispersion, we compared the results of data analysis before and after removing duplicate reads. Deduplication did not reduce QCC values to ~1, and in some cases, led to an increase in QCC (Supplementary Fig. S9) showing that other sources of noise were major contributors to AI overdispersion.

Deduplication can lead to loss of large amounts of legitimate data, and may have other undesirable impacts, such as distorting signal distribution in the biological sample[39–41]. Thus, from a practical standpoint, read deduplication has limited utility, and its impact on AI overdispersion is accounted for in the QCC analysis. Note that in paired-end RNA-seq data, the length of cDNA fragment creates unique molecular identifiers (UMI)[42]. Thus, the results of deduplication in paired-end data (Supplementary Fig. S9d) suggest that the use of UMIs does not remove all AI overdispersion.

Taken together, these observations suggest that library generation is the most likely source of experiment-specific AI overdispersion, while PCR duplicates are at most partially responsible for this technical variability.

## Discussion

We presented analytical and computational analyses showing that, unless very restrictive models of AI signal (or noise) are used, data from a single RNA-seq library is insufficient for reliable quantification of the contribution of technical noise to the observed AI signal (see Fig. 1 and Supplementary Note S1). To gauge variation in experimental noise, we generated 18 RNA-seq libraries prepared from the same RNA using two different protocols and three different starting amounts of total RNA. Analysis of this data and smaller existing datasets showed that technical noise can vary several-fold between experiments, demonstrating that an assumption of a uniform noise model for all RNA-seq experiments would be incorrect.

To account for technical noise and thus enable accurate estimation of AI from RNA-seq data, we developed a computational approach that compares two or more technical replicates and implemented it in a software package, Qllelic (github.com/gimelbrantlab/Qllelic). This approach is conceptually simple; it is equivalent to the binomial test with the number of allelic counts reduced by QCC squared, where QCC is quality correction coefficient calculated from comparison of technical replicates. Despite its simplicity, Qllelic fits the observed data well (see Fig. 2f and goodness-of-fit analyses in Supplementary Figs. S10–S13). Application of Qllelic reduces false positive rates due to technical variation close to zero, while preserving detectable AI signal (see Fig. 4c, d, Supplementary Fig. S11d and Supplementary Note S5).

This approach performed much better than binomial test with correction for multiple hypothesis testing (widely used in published studies, e.g.,[5,18–20]) and methods that incorporate over-dispersion terms into one-replicate analysis[21,22] (Supplementary Fig. S4). It is worth noting that fitting an experiment-wide overdispersion parameter ($\rho$) in a beta-binomial model (e.-g.,[21,26,27]) implies that overdispersion increases with gene coverage (Supplementary Fig. S14). By contrast, we found that in experimental data, overdispersion appears to be near constant across all coverage levels (see Fig. 3d), consistent with the QCC model.

Importantly, use of the QCC/Qllelic approach enables robust differential analysis of AI. It also simplifies such analysis across studies and experiments––QCC values can be precomputed, making differential AI analysis fast and convenient. While our analyses focused on genes, experiment-specific AI overdispersion is evident at single SNP level (Supplementary Fig. S15).

Below, we outline two typical use cases for allele-specific expression analysis using technical replicates for noise correction.

Use case 1 is estimation of AI and confidence intervals in one sample. To be of analytical use, point estimates of AI should be accompanied by the CIs. A worked example (Supplementary Note S6) demonstrates the procedure of allele counting and Qllelic analysis to estimate experimental error using two or more technical RNA-seq replicates. This procedure also incorporates testing of a specific null hypothesis (e.g., $H_0 : AI = 0.5$, $p = 0.05$); Bonferroni correction for the multiple hypothesis testing is applied to the whole list of genes with AI and CI estimated. The tested AI value could be the same for all genes (e.g., $AI = 0.5$) or could be separately specified for each gene.

If more than two technical replicates are available, QCC values are first calculated for all pairwise comparisons (see Methods). This helps identify outlier replicates, if any, which can be removed. The replicate data are pooled to determine point AI estimates, with the mean of all the pairwise QCC values used as the experiment-specific QCC. Note that all replicates should be sampled to the same depth, determined by the replicate with the lowest number of reads; to avoid extrapolation, the safe option is to discard the extra reads from other replicates.

Use case 2 is differential AI analysis between two samples. Knowledge of AI point estimates and the width of the AI distributions enable pairwise differential AI analysis. To test for differential AI at a specified level of confidence, we use the proportional test (see Methods; note that requiring that two CIs do not intersect is a much stricter test than the confidence level of each of the CIs). A worked example comparing two clonal cell lines from $129 \times CastF1$ mice is in Supplementary Note S6. Additional conditions can be applied, such as a minimal AI difference in the point estimates.

Insufficiency of a single technical replicate for noise estimation in AI analyses implies that there is a significant uncertainty regarding interpretation of existing datasets. Very few published RNA-seq studies incorporate any technical replicates. In the Geuvadis study using human cells[16], five samples out of 462 had technical replicates, with pairwise QCC values ranging from 1.04 to 1.21, lowering the number of genes called imbalanced up to 1.5-fold (Supplementary Table S2). Considering that over-dispersion can substantially vary even within a series of replicates (see Experiment 1 in Fig. 2), caution should be exercised when analyzing point AI estimates when QCC cannot be established with certainty, as in any studies without technical replicates.

To illustrate the impact of variation in experiment-specific overdispersion on interpretation of existing datasets, consider data from a randomly chosen individual from the GTEx study[5] (Supplementary Fig. S16). A binomial test, which was performed in that study, assumes $QCC = 1$ and would identify 121 genes with rejected null hypothesis ($AI = 0.5$) for liver and 96 such genes for lung sample. In the absence of replicates, the actual QCC for this experiment is not known, but at $QCC = 2$, there would be 28 and 20 such genes, and at $QCC = 3$, respectively 19 and 11. In other words, the extent of variation in the range that we observed using popular commercial kits for library preparation can affect the results of a simple analysis by an order of magnitude.

We assessed computational and experimental steps to identify possible sources of technical overdispersion. Computational procedures for counting allelic reads can influence analysis both in ways that do not affect overdispersion (e.g., reference bias in mapping), and in ways that increase overdispersion. The counting procedure (see Methods) we use to generate input for Qllelic controls for reference bias by mapping reads to two synthetic parental pseudogenomes with SNP (but not indel) substitutions. Note that this simple procedure assumes all SNPs in a gene produce independent counts, which is incorrect when two or more SNPs are within the same read (or read pair). While this issue has no large effect on the main focus of this work, differential AI analysis and variation in overdispersion between

replicates, we note that allelic counts generated by approaches that aim to address this problem[21,43] can also be used as input for Qllelic.

The experimental process of the sequencing library generation, but not sequencing process itself, appears to be the principal contributor to extra-binomial overdispersion estimated via the QCC value. Intriguingly, computational deduplication of reads (removal of potential PCR amplification artifacts) did not eliminate or sometimes even reduce overdispersion (see Supplementary Fig. S9), leaving open the question of the specific molecular process responsible. While there appear to be systematic differences between protocols (compare Expts. 1, 2, and 3), variation between experiments done with the same protocol can still be substantial (e.g., see outlier in Expt. 1 and differences in QCC in Supplementary Fig. S8b, c). It is thus advisable to have control for each sample.

Some tools for RNA-seq analysis[36–38] can perform haplotype assignment of non-allele-informative reads (i.e., reads not covering a SNP are counted towards one or the other allele). This procedure leads to a large increase in the AI overdispersion (see Supplementary Fig. S6) and should thus be avoided.

Use of biological replicates is much more common in RNA-seq studies, especially when multiple congenic individuals (e.g., mice) are available. Biological replicates can formally be used to calculate QCC values. For example, in a study of allele-specific expression in mouse cells[28], application of Qllelic analysis to the two samples with available biological replicates yielded QCC of 1.51 and 1.56 (Supplementary Table S3). This procedure provides a better account for combined biological and technical variation than pooling or averaging of data across the replicates (see Fig. 2g).

However, such use of biological replicates has significant drawbacks. First, the application of QCC to biological replicates relies on the assumption that the variation between these replicates is uniformly distributed across the transcriptome, as it is for technical replicates. If this assumption is incorrect and there are actual differences between the biological replicates (e.g., a gene shifts AI from 0 to 1), estimates of overdispersion might lead to unpredictable errors. Furthermore, without technical replicates it is impossible to separate biological variation from technical noise.

## Methods

**RNA and library preparation.** Total RNA was isolated using Trizol from a freshly collected kidney tissue of an adult female mouse of 129S1 x Cast/Ei F1 background (F1 breeding was performed at the DFCI mouse facility, with parent animals obtained from the Jackson Laboratories. All animal work was performed under DFCI protocol 09-065, approved by the DFCI Institutional Animal Care and Use Committee. Animals were housed in accordance with Guide for the Care and Use of Laboratory Animals). RNA integrity was assessed using Bioanalyzer, and it was quantified using the Qubit device. Aliquots of this total RNA prep were used to prepare three sets of replicate libraries, all starting with polyA RNA isolation: six libraries with NEBNext kit, starting each with 100ng; six libraries using SMARTseq v4 kit starting with 10 ng RNA; and the same, with 0.1 ng RNA. All libraries were prepared at the DFCI sequencing facility according to the manufacturers' instructions. All sequencing was done on HiSeq 2500 machine at the DFCI sequencing facility.

For data analysis example discussed in Use Case 2, Abelson lymphoblastoid clonal cell lines Abl.1 and Abl.2 of 129S1 × Cast/Ei F1 background[14] were cultured in RPMI medium (Gibco), containing 15% FBS (Sigma), 1X L-Glutamine (Gibco), 1X Penicillin/Streptomycin (Gibco) and 0.1% β-mercaptoethanol (Sigma). Total RNA was extracted from cells using a magnetic bead-based protocol using Sera-Mag SpeedBeads (GE Healthcare). Isolated total RNA was DNase-treated with RQ1 DNase (Promega). RNA sequencing libraries were prepared using SMARTseq v.4 kit (Takara) starting with 10 ng total RNA for each replicate. Sequencing was performed on HiSeq4000 platform at Novogene, Inc.

**Additional data sources.** Geuvadis dataset includes RNA-seq data on LCLs established from 462 individuals from five populations[16]. FASTQ files for paired-end reads (2 × 75 bp) for five individuals (HG00117, HG00355, NA06986, NA19095, NA20527), each with 7 replicates, were downloaded from 1000 Genomes project [ftp.1000genomes.ebi.ac.uk/vol1/ftp/phase3/data/]. Allelic count data

(processed using standard GTEx pipeline) for a randomly selected individual GTEX-11NUK from the Midpoint phase of the GTEx project were downloaded from dbGaP [https://www.ncbi.nlm.nih.gov/projects/gap/cgi-bin/study.cgi?study_id=phs000424.v7.p2]. We also used RNA-seq data from mouse neuronal progenitor cells (GSE54016) [https://www.ncbi.nlm.nih.gov/geo/query/acc.cgi?acc=GSE54016].

**AI estimation pipeline.** AI estimation tools described here are implemented in two parts. Data processing steps from read alignment to allelic counts were based on the ASEReadCounter tool in the GATK pipeline[24]. It was re-implemented using in part Python scripts developed by S. Castel (github.com/secastel/allelecounter), and denoted as ASEReadCounter* (github.com/gimelbrantlab/asereadcounter_star). Calculation of QCC, estimation of confidence intervals and differential AI analysis are implemented in Qllelic tool set (github.com/gimelbrantlab/Qllelic).

*Reference preparation.* Two custom parental genomes ("pseudogenomes"[44,45]; see ASEReadCounter* at github.com/gimelbrantlab/asereadcounter_star) were used for mapping as reference. For 129S1 × Cast/Ei F1 cross mouse samples, alleles are determined with maternal and paternal strain genomes and strain-specific variants; for human data (Geuvadis project[16]) phased SNP variant calls were used. Respective allelic variants from Single Nucleotide Polymorphism database 142 (dbSNP142[46]) or 1000 Genomes Project phase 3 structural variant call-set were inserted into the reference genome (GRCm38.86 or hs37d5, 1000 genomes, phase 2), to obtain a pair of "parental" reference genomes for further analysis (for worked example see Supplementary Note S6). For each organism, we also created a vcf file with one allele considered as a reference (maternal 129S1 or first phased allele) and the other as an alternative allele. Only heterozygous sites were used in the downstream analysis.

*Calculation of allelic counts.* Alignment: RNA-seq reads were aligned with STAR aligner (v.2.5.4a)[47] on each of two pseudogenomes, with default threshold on quality of alignment. Only uniquely aligned reads were taken into further consideration (–outFilterMultimapNmax 1 parameter was applied).

Allele assignment: Reads that were present in only one of the alignments, and reads that had better alignment quality for one of the alignments, were assigned to the corresponding allele read group and marked respectively. The remaining reads (not overlapping heterozygous SNP positions) were not used downstream. This procedure is based on Python scripts by S.Castel.

Read deduplication: When applied, Picard (v.2.8.0; broadinstitute.github.io/picard) MarkDuplicates was used.

Library subsampling: To ensure that all aligned counts belong to similar distributions, BAM files corresponding to the same experiment were subsampled to the same size using a custom bash script with randomness generated using the shuf command.

Allelic counting for SNPs: Given a vcf file with heterozygous positions (discussed under Reference preparation), coverage over each SNP was calculated using samtools mpileup (v.1.3.1) and parsed to obtain the table with allelic counts. This procedure is based on Python scripts by S.Castel.

Allelic counting for genes: All exons belonging to the same gene were merged into a single gene model based on GTF file (RefSeq GTF files, GRCm38.68 and GRCh37.63, were downloaded from Ensemble ftp://ftp.ensembl.org/pub/release-68/gtf/[48]), excluding overlapping regions that belong to multiple genes. Phased allelic counts for all SNPs within the whole gene model were summed:

$$M_g = \sum_{SNP \in g} M_{SNP}$$
$$C_g = \sum_{SNP \in g} M_{SNP} + P_{SNP} \tag{1}$$

Unless specified otherwise, only genes with ≥10 total counts were used for further analysis.

Allelic Imbalance estimates: Estimates for AI for a gene g were obtained as a proportion of maternal gene counts ($M_g$) to total allelic gene counts:

$$AI_g = \frac{M_g}{C_g} \tag{2}$$

*Additional tools for AI calculation.* We used three tools in our comparative analyses: Qllelic (v0.3.2), MBASED (v1.20.0) and GeneiASE (v1.0.1). For uniformity, input for comparisons was pre-processed in the same way for all tools. In case of real data, the same genes were filtered so the data will satisfy all the tools requirements on SNP numbers and SNP coverages (see Supplementary Fig. S4). The default parameters of all the tools were used in analyses (see Supplementary Figs. S4, S7b):

1. One-sample analysis: For Qllelic: default parameters of PerformBinTestAIAnalysisForConditionNPoint() function. For MBASED: runMBASED function with isPhased = TRUE, numSim = 10000, and the rest set to default values. For GeneiASE: default parameters of geneiase -t static

2. Two-sample analysis: For Qllelic: default parameters of PerformBinTestAIAnalysisForTwoConditions() function. For MBASED:

`runMBASED` function with `isPhased = TRUE`, `numSim = 10000`, and the rest set to default values. For Gene*i*ASE: default parameters of `geneiase -t icd`

**Calculation of quality correction constant for 2 replicates**. As gene coverage is an essential parameter of proportional beta-binomial model of allelic imbalance, we started with the standard procedure of splitting genes into bins by coverage to discretize our model.

Bin boundaries were defined as rounded up powers of the base $b = 1.05$: $\bar{C} = \{\lceil b^1 \rceil, \lceil b^2 \rceil, \lceil b^3 \rceil, \ldots \}$. Note that QCC calculations do not strongly depend on the exact bin size, see Supplementary Fig. S7. Each gene $g$ was assigned to a bin according to the mean of its counts $C_{1g}$ and $C_{2g}$ from two technical replicates:

$$\forall g : \frac{C_{1g} + C_{2g}}{2} \in B_i = (\bar{C}_{i-1}, \bar{C}_i] \Rightarrow g \in G_i, \quad (3)$$

then each bin $B_i$, containing set of genes $G_i$, was processed separately.

*Fitting AI distribution as beta-binomial mixture.* To fit the parameters of a mixture of two proportional beta-binomial distributions, representing observed AI from the pooled replicate in each coverage bin $B_i$:

$$a_i \sim \begin{cases} \frac{\text{Beta-Bin}(2 \cdot \widehat{C}_i, \alpha_{1i}, \alpha_{1i})}{2 \cdot \widehat{C}_i}, & \text{with probability } \omega_{1i} \\ \frac{\text{Beta-Bin}(2 \cdot \widehat{C}_i, \alpha_{2i}, \alpha_{2i})}{2 \cdot \widehat{C}_i}, & \text{with probability } \omega_{2i} \end{cases} \quad (4)$$

$$\widehat{C}_i = \sqrt{\bar{C}_{i-1} \cdot \bar{C}_i}$$

$$\omega_{1i} + \omega_{2i} = 1$$

$$\alpha_{1i} > 1, \ \alpha_{2i} \in (0, 1),$$

we use Expectation-maximization (EM) algorithm (see Fig. 3d). Our fitting procedure is similar to the procedure used in the classical Gaussian mixture model[49].

For fitting procedure, we used a threshold on the total allelic gene coverage (50 for mice and 30 for human). All bins that didn't meet the requirement of minimal 40 observations (genes) were also excluded from all stages of QCC-fitting process.

Starting from initials $\omega_{1i}^0 = \omega_{2i}^0 = 0.5$, $\alpha_{1i}^0 = 10$, $\alpha_{2i}^0 = \frac{1}{50}$, and vector of converted allelic imbalance observations $\{\text{AI}_{\theta i}\}_{\theta \in \{1..N_i\}}$, where $N_i$ is number of genes in bin $B_i$:

$$x_{ni} = \text{AI}_{ni} \cdot \widehat{C}_i, \quad (5)$$

we performed iterative EM steps until the difference between parameters of the sequential steps converged (Supplementary Fig. S13).

E-step:

$$\gamma_{nki}^t = \frac{\omega_{ki}^{t-1} \text{BetaBin}(x_{ni} \mid 2\widehat{C}_i, \alpha_{ki}^{t-1}, \alpha_{ki}^{t-1})}{\sum_{j=\{1,2\}} \omega_{ji}^{t-1} \text{BetaBin}(x_{ni} \mid 2\widehat{C}_i, \alpha_{ji}^{t-1}, \alpha_{ji}^{t-1})} \quad (6)$$

for $k \in \{1, 2\}$, $n \in \{1, \ldots, N_i\}$ and $t$ is number of EM step.

M-step: Since we expect $\mu = \widehat{C}_i$ and beta-binomial distributions being symmetric:

$$\omega_{ki}^t = \frac{1}{N_i} \sum_{n=1}^{N_i} \gamma_{nki}^t$$

$$\Sigma_{ki}^t = \frac{\sum_{n=1}^{N_i} \gamma_{nki}^t \cdot (x_{ni} - \widehat{C}_i)^2}{\sum_{n=1}^{N_i} \gamma_{nki}^t} \quad (7)$$

$$\Sigma_{ki}^t = \frac{2\widehat{C}_i \cdot {\alpha_{ki}^t}^2 \cdot (2\alpha_{ki}^t + 2\widehat{C}_i)}{4{\alpha_{ki}^t}^2 \cdot (2\alpha_{ki}^t + 1)} = \frac{4 \cdot \widehat{C}_i \cdot \alpha_{ki}^t + (2\widehat{C}_i)^2}{8 \cdot \alpha_{ki}^t + 4} \Rightarrow \alpha_{ki}^t = \frac{(2\widehat{C}_i)^2 - 4\Sigma_{ki}^t}{8\Sigma_{ki}^t - 4\widehat{C}_i}$$

*Simulation of a pair of replicates.* Using fitted triplet of parameters $\{\omega_{1i}, \alpha_{1i}, \alpha_{2i}\}$, in each bin $B_i$ we generated the weighted mixture of two Beta distributions probabilities $\{p_{\theta i}\}_{\theta \in \{1..5000\}}$, for 5000 "genes":

$$\{p_{\theta i}\}_\theta \sim \begin{cases} \text{Beta}(\alpha_{1i}, \alpha_{1i}), & \text{with probability } \omega_{1i} \\ \text{Beta}(\alpha_{2i}, \alpha_{2i}), & \text{with probability } (1 - \omega_{1i}) \end{cases} \quad (8)$$

Next, for each "gene" a pair of beta-binomial distributed AIs is generated, forming two replicates.

$$\{\xi_{1\theta i}, \xi_{2\theta i}\} \sim \frac{\text{Bin}(\widehat{C}_i, p_{\theta i}, p_{\theta i})}{\widehat{C}_i} \quad (9)$$

The expected AI distribution then can be obtained via subtraction: $\xi_{1\theta i} - \xi_{2\theta i}$.

*Quantile analysis and QCC value.* To quantify the overdispersion, we performed quantile analysis between observed $\Delta$AI distribution (Fig. 3c) and expected $\Delta$AI distribution (Fig. 3e), within the coverage bins. It is a reasonable measure because differences between AI values among replicates generally tend to be symmetric on autosomes in experiments.

For each coverage bin $i$ and a set of quantiles $q \in \{0.2, 0.35, 0.5, 0.65, 0.8, 0.9, 0.95\}$, the ratios of quantiles of observed $\Delta$AI to quantiles of expected $\Delta$AI were calculated: $Q_{q,i}^{\text{obs.}} / Q_{q,i}^{\text{exp.}}$.

Then the obtained ratios were linearly fitted with a constant which we call Quality Correction Constant (QCC), since it reflects the difference between observation and the binomial sampling assumption in the model (see Fig. 3f).

**More than 2 replicates in the analysis**. When more than 2 replicates are used in the analysis, gene counts and AI estimates are obtained from all $M \geq 3$ sampled replicates pooled, and the mean of all pairwise QCCs is used for correction of Confidence Intervals (CI):

$$\text{QCC} = \frac{\sum_{r_i, r_j \in \{1..M\}, r_i \neq r_j} \text{QCC}_{r_i r_j}}{\binom{M}{2}}. \quad (10)$$

Note that before performing this step, it is useful to check if any replicates are outliers, and exclude them from further analysis.

**Adjusting AI confidence intervals**. To apply QCC and adjust CI we use proportional test function `prop.test` from R standard package `stats`, using $\text{QCC}^2$ times less allelic coverage and total coverage values.

The reasoning in choosing this test is as follows: we observe that the quantiles of AI differences are QCC times wider than those from proportional binomial assumption about maternal counts distribution relative to total counts $C_g$. To approximate this property for our distribution we treat gene AI observations as proportions which came from the binomial distribution for $\text{QCC}^2$ times less coverage:

$$\text{AI}_g \sim \frac{\text{QCC}^2 \cdot \text{Bin}(\frac{1}{\text{QCC}^2} \cdot C_g, a_i)}{C_g} = \frac{\text{Bin}(\frac{1}{\text{QCC}^2} \cdot C_g, a_i)}{\frac{1}{\text{QCC}^2} \cdot C_g}. \quad (11)$$

In this approximation, gene counts divided by $\text{QCC}^2$ would be generally not integer, which limits the applicability of binomial test but can be addressed with proportional test which is based on Wilson score intervals.

For the methodological consistency, binomial test is implemented as proportional test function `prop.test` on uncorrected gene count values, due to the coincidence of their approximations on integral values.

**Differential AI analysis**. Accurate accounting for CIs enables differential analysis of gene AI both with point estimates and AI values from different samples.

- The difference of AI estimate from particular proportion value is considered significant if the corresponding CI interval does not cover this value.
- For identifying the differently expressed between two samples we use the same function `prop.test` on the respectively corrected on the QCC values estimates as explained above.

**Reporting summary**. Further information on research design is available in the Nature Research Reporting Summary linked to this article.

## Data availability

Sequencing data deposited in GEO as record GSE143310 (ncbi.nlm.nih.gov/geo/query/acc.cgi?acc=GSE143310). Publicly available datasets used: RNA-seq data for 5 individuals from Geuvadis dataset (HG00117, HG00355, NA06986, NA19095, NA20527) [ftp.1000genomes.ebi.ac.uk/vol1/ftp/phase3/data/]; mouse RNA-seq data (GSE54016) [https://www.ncbi.nlm.nih.gov/geo/query/acc.cgi?acc=GSE54016]. Protected datasets used: allelic count data for GTEX-11NUK individual from GTEx project [https://www.ncbi.nlm.nih.gov/projects/gap/cgi-bin/study.cgi?study_id=phs000424.v7.p2]. Source data are provided with this paper, including gene-level allelic counts for the GTEx sample.

## Code availability

AI estimation tools described here are implemented in two parts. Data processing steps from read alignment to allelic counts were reimplemented as ASEReadCounter* (github.com/gimelbrantlab/asereadcounter_star). Calculation of QCC, estimation of confidence intervals and differential AI analysis are implemented in Qllelic tool set (github.com/gimelbrantlab/Qllelic[50]).

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

## Acknowledgements

We thank Drs. M. Gelfand, A. Gusev, A. Favorov, S. Vigneau, A. Pakharev for useful conversations and Andrew Bortvin for help with writing. Supported by NIH grants HD081675 and GM114864 to AAG.

## Author contributions

S.V., S.G. and A.A.G. conceived this project. S.G. performed the experiments and collected data. A.M. developed the statistical approach, in discussions with S.V., A.A.M., S.R.S. and A.A.G., A.M. and S.V. wrote the code. A.M. and A.A.G. wrote the paper, with all authors contributing to writing and editing. A.M. and S.V. contributed equally. A.M. and A.A.G. are co-corresponding authors.

## Competing interests

Authors declare no competing interests.
