## [Peer Review File · Nature Communications]

Reviewers' comments:

Reviewer #1 (Remarks to the Author):

For the task of estimating allelic imbalance from RNA-seq data, the authors demonstrate that the RNA-seq library construction step adds extra variance that is not accounted for by the commonly used binomial test. This results in a larger number of false positives than expected. The authors devise an approach for measuring this additional variance (overdispersion) using technical replicates, where each technical replicate is a separately constructed library from the same source RNA, and summarizing it in terms of a factor they call the Quality Correction Constant (QCC), which may be used to adjust statistical tests of allelic imbalance. The authors demonstrate that their QCC-based approach can effectively mitigate the effect of this extra variability.

Overall, I find the analysis of technical variation contributed by the library construction step to be a welcome contribution. Very few studies generate multiple libraries from the same RNA sample and thus this new data is quite informative. The QCC procedure, which essentially scales counts down to increase the coefficient of variation for binomially-distributed data, appears to effectively eliminate the false positives in allelic imbalance tests that would normally result from this source of variation. My major concerns with the manuscript are with regard to comparison to existing methods that attempt to account for overdispersion and a lack of assessment of the reduction in power (recall) of the QCC-based procedure to detecting genes with allelic imbalance.

Major comments:

1. There are existing statistical methods for accounting for overdispersion in allele-specific expression analyses. This work only compares with the basic binomial-distribution based test, which assumes no overdispersion. Here are a couple of approaches that the QCC-based approach should be compared to:

Skelly D a., Johansson M, Madeoy J, Wakefield J, Akey JM. A powerful and flexible statistical framework for testing hypotheses of allele-specific gene expression from RNA-seq data. *Genome Res.* 2011. doi:10.1101/gr.119784.110

Mayba O, Gilbert HN, Liu J, Haverty PM, Jhunjunwala S, Jiang Z, et al. MBASED: allele-specific expression detection in cancer tissues and cell lines. *Genome Biol.* 2014;15: 405. doi:10.1186/s13059-014-0405-3

2. Related to #1, a natural alternative to the QCC-based approach would be to estimate the overdispersion parameter of a beta-binomial model from the sorts of technical replicates generated as part of this work. This should at least be discussed, and ideally experimented with as a possible approach.

3. The false positive rate is clearly reduced with the QCC-based approach. However, it is not clear what the cost to power/recall is of this approach, nor how it compares to other approaches (e.g., those mentioned in #1). I'd recommend at least simulation studies be performed to characterize the precision recall tradeoffs here.

4. Fig 1 - I do not agree that this figure shows that the discordant binomial test calls are *not* concentrated along the boundary. The only plot in which the discordant calls are clearly not on the boundary is 1g. Discordant points are on the boundary in 1f, the left plot of 1e and, to a lesser extent, in the right plot of 1e. The highlighted point is not terribly representative.

Minor comments:

5. For the experiments in which reads from one library are divided into half-replicates and then compared, I would only expect overdispersion due to the non-independence of the counts (in contrast to sampling with replacement from that library). As such, overdispersion should be most pronounced for genes with low coverage. I would recommend that the Q ratios be shown across coverage bins in this case (e.g., like Fig 2d).

6. The "distributed as" notation " \sim " in some parts of manuscript are non-standard, or incorrect. For example in Fig 2c, you are not scaling a distribution with a weight (w_{1i}), rather there is first an indicator random variable (let's say C_i) that is distributed according to a Bernoulli distribution with parameter (w_{1i}). Then the count is distributed according to a Beta-Binomial distribution with the parameters selected by the C_i indicator (which component of the mixture that count is coming from).

7. Fig 1 caption for "a,b": The text ": Comparison of AI values [maternal allelic counts/(total allelic counts)]; b: comparison of total allelic counts." seems misplaced

8. Fig 2d: please clarify the difference between a "Q ratio" and the QCC

Reviewer #2 (Remarks to the Author):

Allelic imbalance of read-sequencing data is an incredibly powerful tool to study genetic variants underlying differences in important molecular phenotypes. To confidently identify variants with significant effects, researchers must account for unique noise characteristics associated with allelic imbalance (AI) analysis. Therefore, Mendelevich et al set out to develop a framework for accounting for excess variation found in AI estimates. With their newly developed QCC correction factor, they increased the number of significant overlaps between distinct samples. While the researchers highlight library construction as an important consideration when performing AI analysis, I did not find their reference to existing AI research satisfactory. Thus, I am having difficulty identifying key contributions and interpreting the results of the study. Specific comments to follow:

Major:

1. My primary concern is that it is incorrect to claim that extra-binomial variability of AI estimates attributable to technical factors is unexpected when there are multiple references in the literature to read count (including allele-specific) overdispersion attributable to technical factors.

To address this point, I expect further clarification from the authors or an edit of the title/manuscript -- remove "unexpected". Additionally, it would be helpful to understand how the QCC approach compares to existing methods, both advantages and disadvantages. For example, with AI p-values from existing methods (which are more sophisticated than a binomial test), would there still be as many significant effects that fail to replicate?

2. Relatedly, could the authors clarify or remove the statement on the first page? "Efforts to increase AI estimation accuracy have mostly focused on the data analysis, based on the assumption that consistency in measuring total RNA abundance translates to accurate measurement of each of the alleles separately". I did not find this to be an accurate assessment. For example, Figure 5 from (Castel, 2015) shows that even when you scale for read depth RNA sample concentration can have an effect on AI estimates, which seems to contradict the referenced statement.

Minor:

1. I do not have a sense of whether p-values are well calibrated following QCC correction. I appreciate the relevance of certain panels from Figure 4, but the raw p-values can give a better sense of calibration. In particular, I find quantile-quantile plots (comparing theoretical p-values to observed p-values) to be extremely useful for quickly diagnosing whether the p-values are inflated, conservative or calibrated.

2. All the results are based on AI estimates aggregated by gene. Would these results be comparable if analyzing exon-specific AI values?

3. Looks like the panel a and b descriptions are switched in Figure 1.

4. I do not see any orange line in Figure 2d, but I assume the linear fit is the filled line.

Reviewer #3 (Remarks to the Author):

In this work Mendeleevich et al. explore the reproducibility of allele-specific expression analysis and unexpectedly, find high variability of allelic imbalance in technical replicates. The authors propose a statistical and analytical framework that compensates for overdispersion, which allows for more confident calling of significant allelic imbalance as well as facilitating differential allelic imbalance analysis. They also identify library preparation as the largest likely contributor to the technical variability they observe.

Overall, I found the manuscript to be extremely thorough, well organized, and clearly written. The work is presented in a way that makes it broadly accessible to potential end-users of the tool, while at the same time, providing a more technical audience with the information needed. To my knowledge this is by far the most extensive exploration of the topic, and it is a topic of much interest in the functional genomics community. Furthermore, the authors have made their analysis tool available on Github, where it is well documented, which should facilitate its widespread use.

With respect to the core findings of the work, it has already been well-established that allele-specific expression data is over dispersed compared to what is expected under a binomial distribution, and thus, that a simple binomial test is not appropriate and will lead to a large number of false positives. This has been reviewed and demonstrated using the Geuvadis technical replicates in Castel et al., 2015. This limits the novelty of the findings in this study. Therefore, rather than benchmarking their method against a simple binomial test, the authors need to benchmark against either another beta-binomial based tool, or a simplified beta-binomial test that doesn't perform the unique read depth binning of the QCC method.

Furthermore, to demonstrate the utility of their method, and appeal to a broader readership audience and potential users of the method, the manuscript would benefit by applying it to a data set of biological interest and exploring the results. This is particularly true for the framework for identifying differential allelic imbalance, as this is an analysis where there is much interest and few (if any) existing tools to carry it out.

Finally, most population scale RNA-seq studies, such as Geuvadis and GTEx do not perform replicates, somewhat limiting widespread use of the method. The fact that QCC estimates must be generated for each sample, rather than for a few samples in a study and then broadly applied means it is unlikely to see adoption in this context. However, it will likely see adoption in experimental settings, where replicates are much more common, and also clinical settings. RNA-seq, and in particular allele-specific expression analysis are quickly gaining popularity to assist in the diagnosis of Mendelian disease that is missed by genotyping alone (see Cummings et al., 2017 and Mohamadi et al., 2019). I think the authors should highlight both this drawback and potential use case in their discussion.

Overall, if the above concerns are addressed, I think this work will be of sufficient interest and utility to justify publication, and I applaud the authors for the excellent job they've done.

Specific Comments

- It would be helpful to see quantifications of the patterns seen in fig. 1 d/e/f/g and fig. 3b. The statement that the "discordant binomial calls are not concentrated around the boundary determined by binomial test" after applying the QCC method is hard to tell from visual inspection.
- In fig. 3e would make interpretation easier if concordance percentages were listed, as done in panel a.
- Analysis of GTEx data is described in the manuscript text, but details are missing from the Methods – "2. Additional data sources" section.
- When calculating allelic counts at the gene level the authors merged counts across SNPs. Was care taken to avoid double counting the same RNA-seq read if it overlapped multiple SNPs in a

gene? If not, I don't think anything needs to be redone, but this is something authors should be aware of. Tools (e.g. phASER by Castel et al.) take this into account and can better produce gene level counts.

- Please add an explicit part of the methods that describes where the software is available on Github. I was able to find it by going to the lab page and finding the right repository, but it wasn't extremely clear.

- As an aside, using GTEx it would be interesting to generate QCC estimates using different tissues for the same individual "replicates". Analyses of GTEx ASE data have shown that for the vast majority of genes, allelic imbalance is consistent across tissues (GTEx Consortium, 2019), so it may provide a decent estimate of technical variability, but of course would be overly conservative. However, it could serve to identify a set of genes with strong significance of imbalance across tissues.

Response letter (NCOMMS-20-07031-T)

We thank the reviewers for their helpful comments and suggestions. We have extensively revised the manuscript, starting with the title.

We have added a new section to the Results and corresponding new Figure 1, to emphasize theoretical and computational evidence that data from a single RNA-seq library is insufficient for reliable quantification of the contribution of technical noise to the observed AI signal, unless very restrictive assumptions are made about AI signal and/or noise. These analyses are described in detail in the new Supplementary Note 1.

To clarify our main experimental findings, original Figures 1 and 3 have been redesigned and merged into new Figure 2. This includes a significant redesign and added annotation of the AI “discordance map” (Fig.2a, b, f).

To address specific questions from the reviewers, several supplementary figures and notes (and corresponding sections in the main article) have been added. including comparisons to other tools for estimation of allelic overdispersion. These changes and additions are described in detail in our point-by-point response.

The Discussion section has been extensively revised in order to clarify our main findings and highlight differences in our modeling approach.

Following are point-by-point responses to the reviewers’ comments.

Reviewer 1

Major comments:

1. There are existing statistical methods for accounting for overdispersion in allele-specific expression analyses. This work only compares with the basic binomial-distribution based test, which assumes no overdispersion. Here are a couple of approaches that the QCC-based approach should be compared to:

[Skelly et al] [Mayba O et al]

Response:

We thank the reviewer for these suggestions. We have now incorporated comparisons with additional approaches that assume some allelic overdispersion.

We performed detailed comparisons using both real and simulated data with two approaches that are directly comparable with Qllelic: MBASED [Mayba et al, 2014] and GeneiASE [Edsgard et al., 2016]. Both MBASED and GeneiASE use beta-binomial models to describe allelic signal and allow for differential analysis of AI. Both tools employ more sophisticated approaches than used in the GATK pipeline (which we used) for integration of individual SNP counts into gene-level point estimates of AI and estimate overdispersion from SNP-to-SNP variation within one transcript. Importantly, neither tool has an option of calculation of technical overdispersion relative to beta-binomial distribution.

Application of these tools to real data (**Suppl. Fig.S4**) led to two observations. First, when assessing categorical AI calls from technical replicates, both of these tools show concordance lower than our approach. Second, in differential AI analysis using sets of replicates, both tools produce large numbers of false positives (i.e., confident calls of differential AI resulting from comparison of RNA sequencing from the same RNA). Moreover, the extent of discordance varied depending on the experiment. We thus conclude that these tools do not fully account for experiment-specific overdispersion. More generally, we discuss these and other tools using a beta-binomial model with a single overdispersion parameter for all genes (see response to comment 2).

Note that several other existing tools for allele-specific expression analysis were not directly comparable with Qllelic:

(a) The approach described in [Skelly et al., 2011] calibrates the biallelic signal using genomic DNA sequencing. This data type is not available to us. More generally, our observations suggest that such an approach might be suboptimal, since overdispersion in one experiment (gDNA sequencing) is not necessarily indicative of overdispersion in another experiment (RNA sequencing).

(b) QuASAR [Harvey et al., 2015] determines heterozygosity from RNAseq and removes all SNPs with strong allelic bias in RNAseq (all SNPs except SNPs with AI between ~ 0.25 to ~ 0.75) as likely homozygous. It thus uses a small and nonrandom subset of SNPs used by Qllelic, making these tools not directly comparable.

(c) WASP [van de Geijn et.al, 2015] focuses on the regulatory impact of non-coding SNPs and has no output of the ASE (either point estimates or significance values) of the genes that are the presumable targets of regulation.

2. Related to #1, a natural alternative to the QCC-based approach would be to estimate the overdispersion parameter of a beta-binomial model from the sorts of technical replicates generated as part of this work. This should at least be discussed, and ideally experimented with as a possible approach.

Response:

We discuss this question and perform related analyses in the manuscript. In Discussion, we note:

“This approach performed much better than the binomial test with correction for multiple hypothesis testing [...] (**Fig.S4**), and methods that incorporate overdispersion terms into one-replicate analysis [Mayba, 2014; Edsgard, 2016]. It is worth noting that fitting an experiment-wide overdispersion parameter (ρ) in a beta-binomial model [e.g. Skelly, 2011; Mayba, 2014; van de Geijn, 2015] implies that overdispersion increases with gene coverage (**Suppl. Fig.S14**). By contrast, we found that in experimental data, overdispersion appears to be near constant across all coverage levels (see **Fig.3d**), consistent with the QCC model.”

We also added a new Supplementary Note 1, where (in subsection 1.4) we discuss a general point that it may not be appropriate to assume that any particular model is a universal fit to true AI values for all genes in all experiments.

3. The false positive rate is clearly reduced with the QCC-based approach. However, it is not clear what the cost to power/recall is of this approach, nor how it compares to other approaches (e.g., those mentioned in #1). I'd recommend at least simulation studies be performed to characterize the precision recall tradeoffs here.

Response:

We have addressed this question in two ways. First, in the Results section, the revised manuscript emphasizes that the analysis shown in **Fig.4c,d** indicates a reasonable balance between preserving signal and removing false positives. We have also explored the tradeoff between false positives and signal in **Suppl. Note 5**.

“We then asked whether this decrease in FPs was due to overly conservative noise correction. **Fig.4c,d** shows that the computed QCC value is near the point where FP rate reaches the plateau of 0, but not much higher. This suggests that the precision/recall tradeoff is close to optimal. The tradeoff between false positives and signal in differential AI analysis is explored in detail in **Suppl. Note 5**. There, we construct a measure for the fraction of the detected signal and show that any cost when using QCC-corrected test can be compensated with additional sequencing. By contrast, cost in false positives from lack of adjustment for technical overdispersion cannot be reduced by additional coverage.

We conclude that for differential AI analysis, lack of overdispersion correction is likely to result in a very large number of false positives, while QCC correction removes these very effectively. Our analyses of the real and simulated data also suggest that QCC correction is not dramatically overconservative.”

4. Fig 1 - I do not agree that this figure shows that the discordant binomial test calls are **not** concentrated along the boundary. The only plot in which the discordant calls are clearly not on the boundary is 1g. Discordant points are on the boundary in 1f, the left plot of 1e and, to a lesser extent, in the right plot of 1e. The highlighted point is not terribly representative.

Response:

We thank the reviewer for pointing out unclear phrasing. To clarify the point, we have done a major reorganization and redesign of the display items. We redesigned the plot showing the discordant calls (see **Fig.2a**). Among other changes, the single highlighted point is now replaced with a schematic explanation of the discordance map in **Fig.2**, including several hypothetical genes highlighting different states of concordance/discordance between two replicates. We have combined the previous Fig.3 and Fig.1 into new **Fig.2**. In addition to a greater overall clarity, this allows a direct comparison of the distribution of discordant calls before and after the QCC correction relative to the significance boundary. The discordant calls in the QCC-corrected test (**Fig.2f**) are distributed very close to the boundary, in contrast to the binomial test (in **Fig.2b**).

These changes in figures are reflected in a significant rewriting of the Results sections “Variation in the AI estimates across replicate RNA-seq libraries” and “Application of QCC increases concordance between replicates”.

Minor comments:

5. For the experiments in which reads from one library are divided into half-replicates and then compared, I would only expect overdispersion due to the non-independence of the counts (in contrast to sampling with replacement from that library). As such, overdispersion should be most pronounced for genes with low coverage. I would recommend that the Q ratios be shown across coverage bins in this case (e.g., like Fig 2d).

Response:

This analysis is shown in the new **Suppl. Fig.S1**. Overdispersion appears to be about evenly distributed across coverage bins.

6. The "distributed as" notation " \sim " in some parts of manuscript are non-standard, or incorrect. For example in Fig 2c, you are not scaling a distribution with a weight (w_{1i}), rather there is first an indicator random variable (let's say C_i) that is distributed according to a Bernoulli distribution with parameter (w_{1i}). Then the count is distributed according to a Beta-Binomial distribution with the parameters selected by the C_i indicator (which component of the mixture that count is coming from).

Response:

We have changed the notation in this figure (now **Fig.3**).

7. Fig 1 caption for "a,b": The text ": Comparison of AI values [maternal allelic counts/(total allelic counts)]; b: comparison of total allelic counts." seems misplaced

Response:

We thank the reviewer for pointing out this typo.

8. Fig 2d: please clarify the difference between a "Q ratio" and the QCC

Response:

We clarified in the figure and the text that QCC is a fitted value for quantile ratios across all coverage bins. (Note: This panel is now in **Fig.3f**).

Reviewer 2

Major:

1. My primary concern is that it is incorrect to claim that extra-binomial variability of AI estimates attributable to technical factors is unexpected when there are multiple references in the literature to read count (including allele-specific) overdispersion attributable to technical factors.

To address this point, I expect further clarification from the authors or an edit of the title/manuscript -- remove "unexpected".

Response:

We thank the reviewer for pointing out unclear phrasing. Throughout the revised manuscript, we have emphasized that the existence of read-count (abundance) and allelic overdispersion has been reported.

Additionally, it would be helpful to understand how the QCC approach compares to existing methods, both advantages and disadvantages. For example, with AI p-values from existing methods (which are more sophisticated than a binomial test), would there still be as many significant effects that fail to replicate?

Response:

We thank the reviewer for these suggestions. Our response overlaps with the response to questions 1 and 2 from Reviewer 1:

We have now incorporated comparisons with additional approaches that assume some allelic overdispersion.

We performed detailed comparisons using both real and simulated data with two approaches that are directly comparable with Qllelic: MBASED [Mayba et al, 2014] and GeneiASE [Edsgard et al., 2016]. Both MBASED and GeneiASE use beta-binomial models to describe allelic signal and allow for differential analysis of AI. Both tools employ more sophisticated approaches than used in the GATK pipeline (which we used) for integration of individual SNP counts into gene-level point estimates of AI and estimate overdispersion from SNP-to-SNP variation within one transcript. Importantly, neither tool has an option of calculation of technical overdispersion relative to beta-binomial distribution.

Application of these tools to real data (**Suppl. Fig.S4**) led to two observations. First, when assessing categorical AI calls from technical replicates, both of these tools show concordance lower than our approach. Second, in differential AI analysis using sets of replicates, both tools produce large numbers of false positives (i.e., confident calls of differential AI resulting from comparison of RNA sequencing from the same RNA). Moreover, the extent of discordance varied depending on the experiment. We thus conclude that these tools do not fully account for experiment-specific overdispersion. More generally, we discuss these and other tools using a beta-binomial model with a single overdispersion parameter for all genes (see response to comment 2).

Note that several other existing tools for allele-specific expression analysis were not directly comparable with Qllelic:

(a) The approach described in [Skelly et al., 2011] calibrates the biallelic signal using genomic DNA sequencing. This data type is not available to us. More generally, our observations suggest that such an approach might be suboptimal, since overdispersion in one experiment (gDNA sequencing) is not necessarily indicative of overdispersion in another experiment (RNA sequencing).

(b) QuASAR [Harvey et al., 2015] determines heterozygosity from RNAseq and removes all SNPs with strong allelic bias in RNAseq (all SNPs except SNPs with AI between ~0.25 to ~0.75) as likely homozygous. It thus uses a small and nonrandom subset of SNPs used by Qllelic, making these tools not directly comparable.

(c) WASP [van de Geijn et.al, 2015] focuses on the regulatory impact of non-coding SNPs and has no output of the ASE (either point estimates or significance values) of the genes that are the presumable targets of regulation.

More generally, we discuss these and other tools using a beta-binomial model with a single overdispersion parameter for all genes. In Discussion, we note:

“This approach performed much better than the binomial test with correction for multiple hypothesis testing [...] (**Fig.S4**), and methods that incorporate overdispersion terms into one-replicate analysis [Mayba, 2014; Edsgard, 2016]. It is worth noting that fitting an experiment-wide overdispersion parameter (ρ) in a beta-binomial model [e.g. Skelly, 2011; Mayba, 2014; van de Geijn, 2015] implies that overdispersion increases with gene coverage (**Suppl. Fig.S14**). By contrast, we found that in experimental data, overdispersion appears to be near constant across all coverage levels (see **Fig.3d**), consistent with the QCC model.”

We also added a new Supplementary Note 1, where (in subsection 1.4) we discuss a general point that it may not be appropriate to assume that any particular model is a universal fit to true AI values for all genes in all experiments.

2. Relatedly, could the authors clarify or remove the statement on the first page? “Efforts to increase AI estimation accuracy have mostly focused on the data analysis, based on the assumption that consistency in measuring total RNA abundance translates to accurate measurement of each of the alleles separately”. I did not find this to be an accurate assessment. For example, Figure 5 from (Castel, 2015) shows that even when you scale for read depth RNA sample concentration can have an effect on AI estimates, which seems to contradict the referenced statement.

Response:

In the revised manuscript, this unclear statement has been replaced by more detailed discussions of other tools, including analyses shown in **Suppl. Figures S4, S10, S11, and S12**.

Minor :

1. I do not have a sense of whether p-values are well calibrated following QCC correction. I appreciate the relevance of certain panels from Figure 4, but the raw p-values can give a better sense of calibration. In particular, I find quantile-quantile plots (comparing theoretical p-values to observed p-values) to be extremely useful for quickly diagnosing whether the p-values are inflated, conservative or calibrated.

Response:

We thank the reviewer for the suggestion. In the revised manuscript, QQ plots are shown in **Suppl. Fig.S11**. Overall, our analyses show that QCC correction characterizes experimental data well (see **Suppl. Figures S10-S12**). In addition, we compare overdispersion model in QCC/Qllelic with that of beta-binomial models with a single overdispersion parameter for all genes, as described in the response to comment 1.

We also discuss the question of false positive/signal tradeoff in **Suppl. Note 5**.

2. All the results are based on AI estimates aggregated by gene. Would these results be comparable if analyzing exon-specific AI values?

Response:

To perform a completely disaggregated analysis, we assessed whether overdispersion differs between experiments on single SNP level (**Suppl.Fig.S15**). This analysis shows that differences in overdispersion remain.

3. Looks like the panel a and b descriptions are switched in Figure 1.

4. I do not see any orange line in Figure 2d, but I assume the linear fit is the filled line.

Response:

We thank the reviewer for pointing out these typos.

Reviewer 3

With respect to the core findings of the work, it has already been well-established that allele-specific expression data is over dispersed compared to what is expected under a binomial distribution, and thus, that a simple binomial test is not appropriate and will lead to a large number of false positives. This has been reviewed and demonstrated using the Geuvadis technical replicates in Castel et al., 2015. This limits the novelty of the findings in this study. Therefore, rather than benchmarking their method against a simple binomial test, the authors need to benchmark against either another beta-binomial based tool, or a simplified beta-binomial test that doesn't perform the unique read depth binning of the QCC method.

Response:

In the revised manuscript, we aimed to clarify that our central point is less the existence of allelic overdispersion than the extent of its variation between experiment and the insufficiency of a single technical replicate. We also assessed performance of specific tools and the general issue of using a single overdispersion parameter in beta-binomial models (Reviewer 1 has brought up similar points in questions 1 and 2, and we refer to our extensive responses to those).

The reviewer is correct in pointing out that [Castel et al, 2015] discuss the shortcomings of the binomial test. However, even there, it ended up as the only specifically mentioned statistical test in the recommended data analysis pipeline (see Suppl. Fig.10 in that paper). This underscores the need for a conceptually simple quantitative approach to allele-specific analysis.

Furthermore, to demonstrate the utility of their method, and appeal to a broader readership audience and potential users of the method, the manuscript would benefit by applying it to a data set of biological interest and exploring the results. This is particularly true for the framework for identifying differential allelic imbalance, as this is an analysis where there is much interest and few (if any) existing tools to carry it out.

Response:

We thank the reviewer for these suggestions. In the revised manuscript, we have emphasized the utility of our approach for differential AI studies. We also point out a differential AI analysis on clonal cell lines in a use case in the manuscript and in the instruction manual for the Qllelic package on github. In addition, we describe a re-analysis of existing data on clonal cell populations (**Suppl. Table S3**).

Furthermore, we performed a separate study that focuses on biological impact of drugs on changes in AI, relying on Qllelic pipeline for RNA-seq analysis. That manuscript is currently in review. A copy is appended as an accompanying manuscript; a preprint is also posted on <https://www.biorxiv.org/content/10.1101/2020.02.20.954834v2>.

Finally, most population scale RNA-seq studies, such as Geuvadis and GTEx do not perform replicates, somewhat limiting widespread use of the method. The fact that QCC estimates must be generated for each sample, rather than for a few samples in a study and then broadly applied means it is unlikely to see adoption in this context. However, it will likely see adoption in experimental settings, where replicates are much more common, and also clinical settings. RNA-seq, and in particular allele-specific expression analysis are quickly gaining popularity to assist in the diagnosis of Mendelian disease that is missed by genotyping alone (see Cummings et al., 2017 and Mohamadi et al., 2019). I think the authors should highlight both this drawback and potential use case in their discussion.

Response:

We completely agree that overdispersion estimates should not be generated for a few samples in a study and then broadly applied to other samples in the same study or to other studies. However, note that experiment-specific overdispersion is a property of RNA-seq, and QCC/Qllelic analysis just reveals it. We would argue that replicates are necessary and appropriate controls for quantitative RNA-seq studies of allele-specific expression.

The insufficiency of one technical replicate is an important point and we discuss it in detail in new **Fig.1** and new **Supplementary Note 1**.

Specific Comments

- It would be helpful to see quantifications of the patterns seen in fig. 1 d/e/f/g and fig. 3b. The statement that the “discordant binomial calls are not concentrated around the boundary determined by binomial test” after applying the QCC method is hard to tell from visual inspection.

Response:

We thank the reviewer for pointing out unclear phrasing. To clarify the point, we have done a major reorganization and redesign of the display items. We redesigned the plot showing the discordant calls (see **Fig.2a**). Among other changes, the single highlighted point is now replaced with a schematic explanation of the discordance map in **Fig.2**, including several hypothetical genes highlighting different states of concordance/discordance between two replicates. We have combined the previous Fig.3 and Fig.1 into new **Fig.2**. In addition to a greater overall clarity, this allows a direct comparison of the distribution of discordant calls before and after the QCC correction relative to the significance boundary. The discordant calls in the QCC-corrected test (**Fig.2f**) are distributed very close to the boundary, in contrast to the binomial test (in **Fig.2b**).

These changes in figures are reflected in a significant rewriting of the Results sections “Variation in the AI estimates across replicate RNA-seq libraries” and “Application of QCC increases concordance between replicates”.

Additionally, we extended the theoretical considerations in **Suppl.Note 2** with a figure about the real data, showing how overdispersion affects the distribution of the genes with discordant calls relative to the test boundary.

- In fig. 3e would make interpretation easier if concordance percentages were listed, as done in panel a.

Response:

This panel (now **Fig.2g**) now lists percentages.

- Analysis of GTEx data is described in the manuscript text, but details are missing from the Methods – “2. Additional data sources” section.

Response:

We thank the reviewer for pointing this out. This description has now been moved to Methods.

- When calculating allelic counts at the gene level the authors merged counts across SNPs. Was care taken to avoid double counting the same RNA-seq read if it overlapped multiple SNPs in a gene? If not, I don't think anything needs to be redone, but this is something authors should be aware of. Tools (e.g. phASER by Castel et al.) take this into account and can better produce gene level counts.

Response:

We added the following section in the Discussion:

“Computational procedures for counting allelic reads can influence analysis both in ways that do not affect overdispersion (e.g., reference bias in mapping), and in ways that increase overdispersion. The counting procedure (see Methods) we use to generate input for Qllelic controls for reference bias by mapping reads to two synthetic parental pseudogenomes with SNP (but not indel) substitutions. Note that this simple procedure assumes all SNPs in a gene produce independent counts, which is incorrect when two or more SNPs are within the same read (or read pair). While this issue has no large effect on the main focus of this work, differential AI analysis and variation in overdispersion between replicates, we note that allelic counts generated by approaches that aim to address this problem [Mayba, 2011; Castel, 2016] can also be used as input for Qllelic.”

- Please add an explicit part of the methods that describes where the software is available on Github. I was able to find it by going to the lab page and finding the right repository, but it wasn't extremely clear.

Response:

We thank the reviewer for pointing out this omission. Now the github URL is listed in the main text and in Methods.

- As an aside, using GTEx it would be interesting to generate QCC estimates using different tissues for the same individual “replicates”. Analyses of GTEx ASE data have shown that for the vast majority of genes, allelic imbalance is

consistent across tissues (GTEx Consortium, 2019), so it may provide a decent estimate of technical variability, but of course would be overly conservative. However, it could serve to identify a set of genes with strong significance of imbalance across tissues.

Response:

This is an interesting idea to explore. Such an approach could serve at the very least to identify outlier samples (due to either distinct regulatory patterns or to technical issues). A detailed study would be needed to understand how a QCC-like approach should be applied to samples with disparate expression profiles.

Reviewer #1 (Remarks to the Author):

With this revision, the authors have addressed much of my prior comments. I have a few remaining comments:

Major:

1. I appreciate the addition of the power analysis (Supp Note 5), however I found this section to be challenging to digest and the takeaway message regarding how much power is sacrificed for the reduction in false positive rate by QCC was unclear. For example, it is unclear what fraction of genes had differential AI (what fraction belonged to H1 vs H0) and whether this parameter is important. It was also unclear if the QCC-based method was estimating the QCC from the simulated data, or if the QCC used in simulation was given to the QCC-based method (which would not be fair - one does not know the QCC ahead of time). The comparison to the simple binomial test is also not sufficient given that there are other more advanced methods. Including MBASED in this analysis, for example, seems warranted. Perhaps the simulation experiments can be simplified to the case of detecting $AI \neq 0.5$ in a single sample to make things easier to digest. I suggest that the power analysis results be presented as part of the main text, as reduction of false positives is only interesting if power is not significantly reduced. The reduction in power shown in Figure SN 5.4 (middle and right panel) seems a bit worrisome.

2. The conclusion that AI overdispersion is "experiment-specific" does not seem well supported by the results or is too imprecise of a statement. In this work, each "experiment" had a different library protocol or input amount. This work does not demonstrate that AI overdispersion varies across experiments in which the library protocol and input amount is constant, as would be common within a single study. Thus, it does not preclude the possibility of using technical replicates for just one experiment in the study and then using the overdispersion estimated from that experiment for all others in that study, assuming library protocols and input amounts are kept constant.

Minor:

3. Given the questions regarding the source of technical variation, it is important that the authors clearly define what is meant by a "technical replicate" early on in the manuscript. In some RNA-seq studies, technical replicate can simply refer to sequencing the same library across multiple lanes on a sequencer, or in multiple runs.

4. Can the method presented be interpreted as one that estimates a different dispersion parameter for the beta-binomial distribution within each coverage bin, rather than a single dispersion parameter across all genes? If so, why use QCC instead of simply using beta-binomial tests within each bin with a bin-specific dispersion parameter?

Reviewer #2 (Remarks to the Author):

My concerns have been addressed.

Reviewer #3 (Remarks to the Author):

The authors have addressed my concerns in their revised manuscript. However, I do take significant issue with their revised title: "Quantification of allelic imbalance requires replicate sequencing libraries". I appreciate that they changed the previous title to address other concerns, but I do not believe the revised title is warranted based on the findings.

As the authors point out in their introduction "a near-universal practice in such studies is to prepare and sequence only one library per RNA sample". This is likely not going to change for population-scale transcriptomic samples where tens of thousands of samples are being sequenced.

However, I do agree with the authors' point that in experimental settings (and wherever else it is financially feasible) replicates should be performed.

My concern with the title is that it would do damage to the field by creating misunderstanding. To date, the majority of allelic imbalance studies have not had replicates, and yet have made important and replicable scientific findings. In light of the authors' work, it's likely that some of those findings may be false positives, but they are clearly not all invalidated.

Thus, I would suggest the authors tone down the title. Something along the lines of "dramatically improve" vs "require".

Response letter (NCOMMS-20-07031B)

We thank the reviewers for their helpful comments and suggestions. Following are point-by-point responses to the reviewers' comments.

Reviewer #1:

Major:

1. I appreciate the addition of the power analysis (Supp Note 5), however I found this section to be challenging to digest and the takeaway message regarding how much power is sacrificed for the reduction in false positive rate by QCC was unclear. For example, it is unclear what fraction of genes had differential AI (what fraction belonged to $H1$ vs $H0$) and whether this parameter is important.

Response:

This is an important point that we have now made more clear in the first paragraph of Supplementary Note 5:

“We showed that QCC correction dramatically lowers false positive (FP) rate. A natural question is what is the trade-off in terms of the loss of signal. Note that "true positive" is more challenging to determine than FP. An intuition can be given by pixels forming an image: with unlimited resolution, any two distinct dots can be resolved. However, any specific resolution imposes a limit on the ability to resolve dots. Library complexity is equivalent to resolution while coverage depth is equivalent to magnification. In other words, whether two AI values are distinguishable is a property of the data, and power analysis is an estimation of necessary coverage for a particular needed level of resolution with a given quality of the data.”

Even in idealized conditions when we know all true AI values, there is no predetermined set of “genes with differential AI” which is independent of arbitrary threshold on delta AI (resolution) in addition to coverage.

Furthermore, in realistic experimental conditions, “true” AI values are not known. Depending on data quality, the achievable resolution is always limited. There is, thus, no “sacrifice” of true positives, when using the test which is appropriate to the data.

It was also unclear if the QCC-based method was estimating the QCC from the simulated data, or if the QCC used in simulation was given to the QCC-based method (which would not be fair - one does not know the QCC ahead of time).

The comparison to the simple binomial test is also not sufficient given that there are other more advanced methods. Including MBASED in this analysis, for example, seems warranted. Perhaps the simulation experiments can be simplified to the case of detecting $AI \neq 0.5$ in a single sample to make things easier to digest.

Response:

Suppl. Note 5 is not focused on method comparisons, but rather on the demonstration of the effect of underestimation of overdispersion in principle. We performed comparisons of MBASED and GeneiASE elsewhere (Suppl.Fig.S7 on simulations and Suppl.Fig.S4 on real data); we feel that adding them here would distract from the main point.

We have added clarification to Suppl.Fig.S7 that QCC was computed there from scratch.

I suggest that the power analysis results be presented as part of the main text, as reduction of false positives is only interesting if power is not significantly reduced. The reduction in power shown in Figure SN 5.4 (middle and right panel) seems a bit worrisome.

Response:

We agree with the reviewer that the power analysis is important. We found it difficult to condense the content of the Suppl. Note 5 sufficiently for insertion in the main text. We thus somewhat expanded the Note to clarify it, and emphasized the discussion of power analysis in the main text as follows:

“In that Note, we perform power analysis and show that any cost when using QCC-corrected test can be compensated with additional sequencing. By contrast, cost in false positives from lack of adjustment for technical overdispersion cannot be reduced by additional coverage.”

2. The conclusion that AI overdispersion is "experiment-specific" does not seem well supported by the results or is too imprecise of a statement. In this work, each "experiment" had a different library protocol or input amount. This work does not demonstrate that AI overdispersion varies across experiments in which the library protocol and input amount is constant, as would be common within a single study. Thus, it does not preclude the possibility of using technical replicates for just one experiment in the study and then using the overdispersion estimated from that experiment for all others in that study, assuming library protocols and input amounts are kept constant.

Response:

We thank the reviewer for pointing this out. We added the following clarification to Discussion:

“While there appear to be systematic differences between protocols (compare Expts. 1, 2 and 3), variation between experiments done with the same protocol can still be substantial (e.g., see outlier in Expt 1 and differences in QCC in Suppl Fig 8b,c). It is thus advisable to have control for each sample.”

Minor:

3. Given the questions regarding the source of technical variation, it is important that the authors clearly define what is meant by a "technical replicate" early on in the manuscript. In some RNA-seq studies, technical replicate can simply refer to sequencing the same library across multiple lanes on a sequencer, or in multiple runs.

Response:

We placed the definition in the second paragraph in the Introduction.

4. Can the method presented be interpreted as one that estimates a different dispersion parameter for the beta-binomial distribution within each coverage bin, rather than a single dispersion parameter across all genes? If so, why use QCC instead of simply using beta-binomial tests within each bin with a bin-specific dispersion parameter?

Response:

Bin-specific correction would be less precise since the aim of binning is to estimate a statistic based on multiple observations. In addition, we believe that one of the advantages of the QCC approach is a conceptual simplicity of using a single genome-wide parameter. It is also computationally simpler to use a universal genome-wide correction on single value, even though it is possible to, e.g., recalculate QCC into rho values for each particular gene (as we show in Suppl. Fig.S14).

Reviewer #3:

The authors have addressed my concerns in their revised manuscript. However, I do take significant issue with their revised title: "Quantification of allelic imbalance requires replicate sequencing libraries". I appreciate that they changed the previous title to address other concerns, but I do not believe the revised title is warranted based on the findings.

As the authors point out in their introduction "a near-universal practice in such studies is to prepare and sequence only one library per RNA sample". This is likely not going to change for population-scale transcriptomic samples where tens of thousands of samples are being sequenced. However, I do agree with the authors' point that in experimental settings (and wherever else it is financially feasible) replicates should be performed.

My concern with the title is that it would do damage to the field by creating misunderstanding. To date, the majority of allelic imbalance studies have not had replicates, and yet have made important and replicable scientific findings. In light of the authors' work, it's likely that some of those findings may be false positives, but they are clearly not all invalidated.

Thus, I would suggest the authors tone down the title. Something along the lines of "dramatically improve" vs "require".

Response:

We respectfully disagree. We feel that the field would greatly benefit from a conversation about limitations in study design and analysis, and our paper, including its title, aims to be a part of such a conversation. There are examples of large studies of allele-specific expression whose statistical deficiencies have rendered them completely useless (for example, see B. DeVeale, D. van der Kooy, T. Babak, Critical evaluation of imprinted gene expression by RNA-Seq: a new perspective. PLoS genetics 8, e1002600 (2012)). Most studies of course exceed such a low standard, but we should aim for as much clarity as possible. At the very minimum, for studies with single replicates (such as existing datasets) it is worth emphasizing the extent of uncertainty to aid interpretation of results and help guide validation follow-up studies.

Reviewer #1 (Remarks to the Author):

With this revision the authors have largely addressed my previous concerns. Regarding the appropriateness of the title (brought up by reviewer #3) and my prior comment about the conclusion of AI overdispersion being "experiment-specific" not being well supported: I still think that the conclusion that replicates are absolutely required for every single sample in a study is too strong. What is clear to me from the results is that there is a library-protocol effect and some protocols are much better than others. In particular, it appears that the protocol for Experiment 2 (SMARTseq v4, 10 ng) is better suited for this type of analysis, yielding concordance values that are close to what one would get with a binomial relationship. Having one outlier sample in Experiment 1 is not really convincing that the QCC varies significantly from sample to sample for a fixed protocol. The take home message that I get is that one needs to be careful about the RNA-seq protocol used for allelic imbalance studies. Follow-up work could be done to identify the protocol that results in a QCC value closest to one and that should be recommended to the community. I would be more accepting of the title if the authors provided some more practical guidance in the discussion. For example, having replicates for every experiment is best, but if that is not possible, choose protocol X because it resulted in the smallest QCC in this study, and perhaps generate replicates for a few random experiments to get an estimate of the QCC.